# Multi-Agent Design: Optimizing Agents with Better Prompts and Topologies

**Han Zhou**[1,3,*]  **Xingchen Wan**[2]  **Ruoxi Sun**[2]  **Hamid Palangi**[1]  **Shariq Iqbal**[2]
**Ivan Vulić**[2,3]  **Anna Korhonen**[3]  **Sercan Ö. Arık**[1]
[1]Google  [2]Google DeepMind  [3]University of Cambridge
{hzhouml,soarik}@google.com

## Abstract

Large language models, employed as multiple agents that interact and collaborate with each other, have excelled at solving complex tasks. The agents are programmed with *prompts* that declare their functionality, along with the *topologies* that orchestrate interactions across agents. Designing prompts and topologies for multi-agent systems (MAS) is inherently complex. To automate the entire design process, we first conduct an in-depth analysis of the design space aiming to understand the factors behind building effective MAS. We reveal that prompts together with topologies play critical roles in enabling more effective MAS design. Based on the insights, we propose Multi-Agent System Search (Mass), a MAS optimization framework that efficiently exploits the complex MAS design space by interleaving its optimization stages, from local to global, from prompts to topologies, over three stages: 1) block-level (*local*) prompt optimization; 2) workflow topology optimization; 3) workflow-level (*global*) prompt optimization, where each stage is conditioned on the iteratively optimized prompts/topologies from former stages. We show that Mass-optimized multi-agent systems outperform a spectrum of existing alternatives by a substantial margin. Based on the Mass-found systems, we finally propose design principles behind building effective multi-agent systems.

## 1 Introduction

Large language models (LLMs) have showcased extraordinary capabilities in understanding, reasoning, and generating coherent responses based on user prompts, revolutionizing a wide range of applications (Ouyang et al., 2022; Kojima et al., 2022). LLM-based agents enhance usability by autonomously handling complex tasks across diverse domains, including code generation and debugging (Jimenez et al., 2023), retrieval-augmented generation (Singh et al., 2025; Wang et al., 2024a), data analysis (Hu et al., 2024b; Guo et al., 2024), and interactive decision-making (Su et al., 2025; Li et al., 2025). These agents are typically programmed with prompts that reinforce them to interact with the environment, utilizing available tools, and approach their objectives over multiple turns (Yao et al., 2023). Beyond individual agents, LLMs can be orchestrated within complex topologies that coordinate multiple agents toward

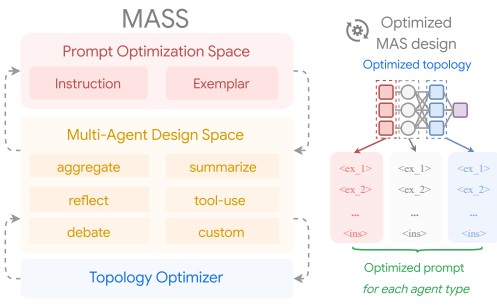

Figure 1: Proposed Multi-Agent System Search (Mass) framework discovers effective multi-agent system designs (with both optimized *topology* and optimized *prompts*, **right**) via interleaved prompt optimization and topology optimization in a customizable multi-agent design space (key components illustrated on the **left**).

a shared objective. This type of multi-agent system (MAS) typically outperforms its single-agent counterpart by involving more diverse agentic perspectives or role profiles, such as agents as verifiers (Shinn et al., 2024) and multi-agent debate (Wang et al., 2024b; Qian et al., 2024).

---

*Work done as a Student Researcher at Google.

However, designing effective MAS for new domains often proves to be challenging. First, the single agent might suffer from prompt sensitivity (Verma et al., 2024), where simple modifications in the prompt can already exert significant but unexpected degradation of performance (Zhou et al., 2024b; Liu et al., 2024a). In MAS, when sensitive agents are cascaded, the compounding effect due to prompt sensitivity may be amplified. Together with the prompt design, crafting an effective topology might demand a substantial amount of manual experimentation, based on trial and error. The problem complexity is exacerbated by the overall combinatorial search space, over not only the unbounded space of prompt design but also the design decisions of what agent to integrate into the topology.

Although recent research has explored automating various aspects of agentic designs, there is still a gap in understanding of *what matters most regarding improved MAS performance*. For example, DSPy (Khattab et al., 2024) automates the process of designing exemplars for improved prompt programming. Li et al. (2024a) propose to optimize MAS by scaling up the number of agents in majority voting. ADAS (Hu et al., 2024a) programs new topologies expressed in code via an LLM-based meta-agent. AFlow (Zhang et al., 2024b) searches better topologies using Monte-Carlo Tree Search within a set of predefined operators. However, the interplay between multiple design spaces, including prompts and topologies, remains unclear.

In this paper, we first conduct in-depth analyses of common design spaces in MAS, examining the impact of aspects such as optimizing the prompts, scaling the number of agents, and involving different types of topologies. Our analyses reveal that prompts frequently form an influential design component that yields strong-performing MAS, and influential topologies only represent a small fraction of the full search space. Based on these insights, we aim to distill the essence of influential MAS components into a pruned search space, thereby lowering the complexity of the overall search process. We propose Multi-Agent System Search (MASS), a novel multi-stage optimization framework that automates the optimization for MAS over an efficient search space. MASS integrates a plug-and-play prompt optimizer and workflow optimizer over a configurable topology space. It overcomes the complexity of joint optimization on MAS by interleaving the optimization stages, from local to global, from prompts to topologies, over three stages: 1) block-level (*local*) prompt 'warm-up' for each topology block; 2) workflow topology optimization in a *pruned* set of topology space; 3) workflow-level (*global*) prompt optimization given the best-found topology.

By optimizing over the identified influential components, MASS yields optimized MAS that achieves state-of-the-art performance, outperforming existing manually-crafted MAS baselines and automatically-generated alternatives, by a substantial margin, demonstrated across an extensive selection of tasks, including reasoning, multi-hop understanding, and code generation. Based on the strongest MAS found by MASS, we provide further insights and guidelines behind building effective MAS. Overall, our contributions can be summarized as follows: 1) we provide an in-depth analysis of the design factors that influence the performance of LLM-based MAS, highlighting the importance of prompts and identifying the influential topologies; 2) we propose MASS, a novel multi-stage optimizer that automates the MAS design by interleaving the optimization of prompts and topologies in an influential search space; 3) MASS shows significant performance improvement on various evaluation benchmarks, delivering guidelines for building effective multi-agent systems for the future.

## 2 DESIGNING MULTI-AGENT SYSTEMS

In this section, we provide a formulation for designing MAS, followed by analyzing the influence of prompt and topology designs. We refer to the structural arrangements of agents as the *topology* of agents and define *workflow* as the logical sequence across different topologies that builds the MAS. The design of a MAS can thus be broadly divided into two levels: block-level design and workflow-level orchestration. At the block level, a *building block* infers a group of agents that work together for a certain function (e.g., `debate`), and we aim to design effective individual agents that best perform their intended role with better *prompt* design. On the other hand, at the workflow level, the optimization involves determining the types and quantities of agents to include and how to arrange them in the most effective way, referred to as the topology optimization. Formally, given a search space $\mathcal{A} = \{a_i\}$ that defines all valid agentic configuration over the building block search dimensions $a_i$ (see Fig. 3), individual agents and their composite topologies can both be parameterized as specific configurations in this search space. To obtain the best workflow $\mathcal{W}^*(a)$, the *workflow topology optimization* is framed as the following optimization problem with an objective function $f(\cdot, \cdot)$ on a

target input and output set $(x, y) \sim \mathcal{D}$ in the defined search space $\mathcal{A}$:

$$\mathcal{W}^*(a) = \arg\max_{a \sim \mathcal{A}} \mathbb{E}_{(x,y) \sim \mathcal{D}}[f(\mathcal{W}(a(x)), y)]. \tag{1}$$

In the rest of this section, we provide an in-depth analysis of each component of MAS design.

## 2.1 BLOCK-LEVEL ANALYSIS: PROMPT DESIGN FOR AGENTS

At the block level, the primary "optimizable component" that significantly influences downstream performance is the *prompt*, which defines the role of the agent (e.g., "*You are an expert in reflecting on errors...*"), provides additional instructions to shape its behavior (e.g., "*You should think step by step...*") and optionally, contains *few-shot demonstrations* (in-context examples) to guide the agent's responses (Wan et al., 2024). For instance, a state-of-the-art prompt optimizer searches both instructions and few-shot demonstrations, where demonstrations are bootstrapped from the model's own, correct predictions on the validation set based on a validation metric. Conditioned on the demonstrations, the prompt optimizer then proposes a few candidates for the instruction with a dataset summary or various hints to improve candidate diversity (Opsahl-Ong et al., 2024). The instructions and demonstrations are then jointly optimized.

Although it is well known that LLMs are sensitive to prompts (Zhou et al., 2024a; Verma et al., 2024), applying automatic prompt optimization (APO) techniques to MAS is not straightforward. In single-turn tasks, APO can be easily performed by treating prompts as optimizable variables and performance over a validation set as the target. However, in MAS, APO becomes more complex due to the interdependence across agents (e.g., the output of one agent may be the input of another agent in a cascade with ground-truth responses for intermediate outputs not being available) and exponentially increasing complexity for combinatorial optimization with more number of agents $n$ involved; The reward signals also become more sparse when $n$ increases, preventing us for implementing APO directly on MAS in any manageable budget; as such, many prior works (Zhang et al., 2024f; Xia et al., 2024) in MAS still primarily use hand-crafted prompts instead of including the prompts as optimizable components in the MAS design.

To systematically understand the influence of prompt design in MAS, we specifically and quantitatively analyze the effect of prompt optimization and compare its effectiveness to other operations common in MAS literature, such as scaling with more agents but with default prompts. We conduct APO on a chain-of-thought (Kojima et al., 2022) agent via a state-of-the art prompt optimizer MIPRO (Opsahl-Ong et al., 2024) that is capable of joint instruction and (1-shot) exemplar optimization, and fairly compare the total inference token cost with self-consistency (Kojima et al., 2022), self-refine (Madaan et al., 2024), and multi-agent debate (Du et al., 2024), where the specifications are provided in App. §C. In Fig. 2, prompting, which equips agents with more informative instructions and exemplars, demonstrates significant advantages in its token-effectiveness over other building blocks. Furthermore, by applying self-consistency on top of the prompt-optimized agent, we observe an improved scaling performance on the token cost, whereas standard approaches in scaling the number of agents (e.g. SC,

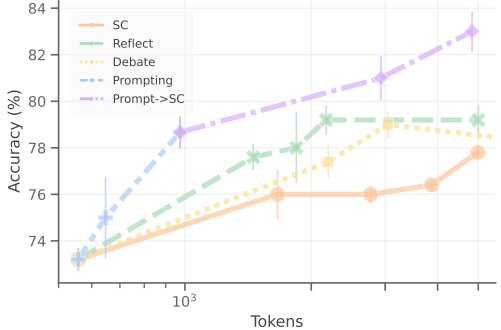

Figure 2: Accuracy vs. total token counts for prompt-optimized agents per question on MATH by Gemini 1.5 Pro compared to scaling agents with self-consistency (SC), self-refine (reflect), and multi-agent debate (debate) only. The error bar indicates 1 standard deviation. We show that by utilizing more compute, better accuracy can be obtained via more effective prompting.

or Reflect) saturate much earlier. This empirical observation sheds light on the importance of prompting while providing early evidence for designing effective MAS – *optimize agents locally before scaling their topology.*

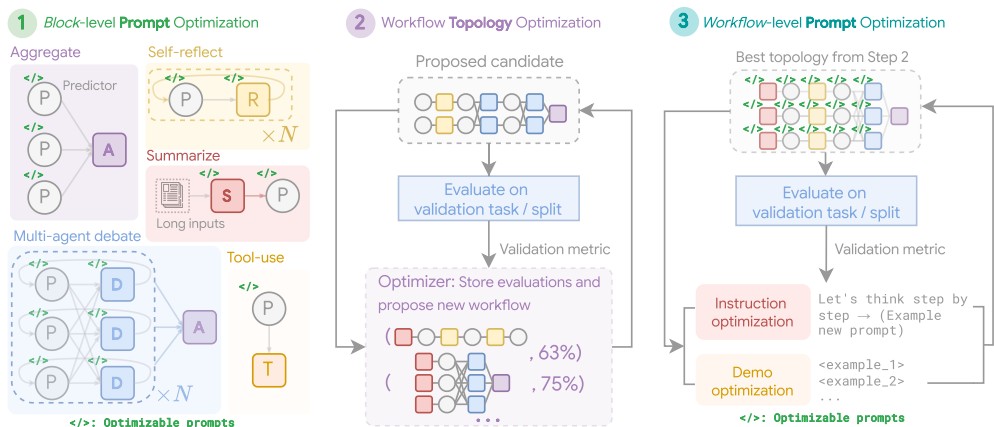

Figure 3: Illustration of the MASS framework with its search space and the optimization. The search space combines both prompts (`Instruction`, `Demo`) and configurable agentic building blocks (`Aggregate`, `Reflect`, `Debate`, `Summarize`, and `Tool-use`). [**1PO**: Block-level **Prompt** Optimization]: we conduct *block*-level prompt optimization for each agentic module individually (denoted by `</>`); [**2TO**: Workflow **Topology** Optimization]: conditioned on the best prompts found in Stage 1 on each agent block, MASS samples valid configurations from an influence-weighted design space while fusing the prompts of each building block from Stage 1; [**3PO**: Workflow-level **Prompt** Optimization]: conditioned on the best workflow found, we again conduct *workflow*-level prompt optimization on the best-found MAS (topologies visualized *for illustration only*).

## 2.2 WORKFLOW-LEVEL SEARCH SPACE DESIGN

At the workflow level, the primary focus is on orchestrating agents to achieve the best performance effectively, and topology optimization has recently garnered significant attention (Li et al., 2024c; Zhang et al., 2024b). However, while much of the existing research emphasizes *search methods*—such as discovering the most efficient and effective way to identify the optimal configuration—there has been less focus on the design of *search spaces*, which determines the perimeter and the scope of any search algorithm. This imbalance draws a parallel to the historical development of *neural architecture search* (NAS) (White et al., 2023). Initially, the field concentrated on sophisticated search methods, such as Bayesian optimization (Kandasamy et al., 2018; Ru et al., 2021) and differentiable search (Liu et al., 2018). Follow-up works have highlighted the often-overlooked importance of search space design, arguing that it can be equally, if not more, critical (Wan et al., 2022; Zhou et al., 2023a).

Inspired by this insight, we hypothesize that manually crafted topologies might be sub-optimal, and automatic topology optimization (potentially framed as a rigorous optimization problem) can play a similarly pivotal role via judiciously designing search space for MAS. To achieve so, we first define an expressive search space, similar to prior works, that consists of the connections between the following *building blocks*, which form a unified search space for MASS:

• *Aggregate*: Agents can collaborate in parallel with diversified predictions, which is then followed by an aggregation operator that obtains the most consistent prediction. The `aggregate` block can be parameterized by $N_a$ agents and also defines the number of agent chains acting in parallel. Majority vote (Li et al., 2024a) and self-consistency (Chen et al., 2024c) sits within this topology.

• *Reflect*: Agents can act as verifiers, providing critics and improvement suggestions based on former predictions. The feedback is then fed into the predictor or the reflector itself for an iterative improvement. Similarly, `reflect` can be parameterized by $N_r$ that defines the number of rounds for self-reflection. The self-refine (Madaan et al., 2024) and Reflexion (Shinn et al., 2024) represent this block, and $N_r$ can be interpreted as a parameter that contributes to the depth of the MAS network.

• *Debate*: Agents in debate can elicit more truthful predictions than a single agent (Du et al., 2024; Liang et al., 2024), where each debating agent would collect opinions from all other agents and provide an updated response. This topology involves a mixture of agents, and $N_d$ defines the number of rounds for debating (i.e., number of fully-connected agent layers in the topology space).

• *Custom Agents*: While the former three forms of agents represent the vast majority of agent topologies constructed as multiple parallel, serial, and a mixture of agents, more versatile definitions

of agents can be inserted into the MAS design space. For example, for task-specific use cases, we introduce an agent as `summarize` to improve the long-context capability in the customizable design space, where $N_s$ defines the rounds of abstraction of information.

• *Tool-use*: Building towards an effective MAS, enabling agents to leverage tools to access external information is critical for system performance, such as using a retriever for RAG (Lewis et al., 2020) and executor with test cases in coding (Chen et al., 2024d). We introduce tool-use (e.g., code `execute`) as an optimizable binary 'insertion' decision $N_T \in \{0, 1\}$ with the predictor.

## 3 MASS: MULTI-AGENT SYSTEM SEARCH

Our analyses in Sec. 2 underscore the importance of well-designed prompts for individual agents and the careful definition of the search space to achieve effective MAS performance. Building on these, we propose a multistage optimization algorithm, **Multi-Agent System Search** (MASS), that surpasses prior arts that focused solely on optimizing workflow topology without appropriate prompt designs. Instead, our approach demonstrates the greater effectiveness of MAS design with properly optimized prompts and thoughtfully designed search spaces. MASS framework is illustrated in Algorithm 1 and Fig. 3, following an intuition from local to global, from block-level to workflow-level, that conquers the complexity of combinatorial optimization with effective per-stage optimization detailed below.

**1) Block-level prompt optimization.** Before composing agents, we first ensure that individual agents are thoroughly optimized at the block level, as highlighted in Sec. 2.1 and Fig. 2 – this step ensures that each agent is primed for its role with the most effective instructions in the most manageable computation budget. To further overcome the complexity of joint optimization on a large MAS space, we first warm up the initial predictor with single-agent APO, $a_0^* \leftarrow \mathcal{O}_\mathcal{D}(a_0)$, where both instruction and exemplars are jointly optimized with the modular prompt optimizer $\mathcal{O}$. Followed by conditioning on the warmed predictor, we continue optimizing each topology with a minimum number of agents, $a_i^* \leftarrow \mathcal{O}_\mathcal{D}(a_i|a_0^*)$, such that, 2 predictors paired with 1 debator form the minimum building block as the `debate` topology, thereby lowering the complexity for optimization, and this topology can be scaled up later with more predictors and debators but all equipped with optimized prompts. To measure the influence of each building block, we store the validation performance once the optimization is completed. It is important that though Stage (1) serves as the *warm-up* stage per building block, it is still a critical stage that guarantees the follow-up topology optimization is searching in an effective space, composing well-performing agents instead of suffering from the compounding impact from any ill-formed agents with manual prompts.

**2) Workflow topology optimization.** In this stage, we focus on optimizing the overall MAS structure, determining the most effective arrangement and connectivity between agents. To understand the influence of individual blocks, we report the performance of various topologies in Fig. 4. It is noticeable that not all topologies are beneficial to MAS design, whereas positively influenced topologies only represent a small fraction of the overall set, such that, in HotpotQA (Yang et al., 2018), only `debate` brings 3% gain while others fail to improve or even degrade systematic performance. We again observe similar trends in the test-output-prediction subtask of LiveCodeBench (Jain et al., 2024). It highlights the importance of searching in the influential set of the search space, whereas including decremental building blocks

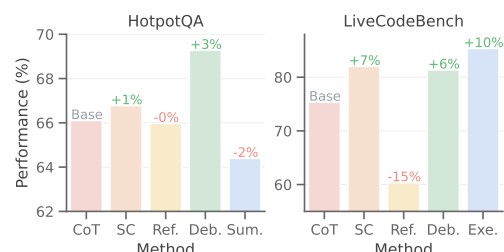

Figure 4: The performance of different topologies with Gemini 1.5 Pro compared to the base agent with each topology being optimized with APO, where `Sum.` (Summarize) and `Exe.` (Executor) are task-specific topologies as illustrated in Fig. 3. We observe that not all topologies have a positive influence on the MAS design.

may not only result in higher search complexity but also degrade the performance. The analysis in Fig. 4 shows that beneficial topologies only represent a small fraction of the full design space. Therefore, we aim to distill the essence of strong-performing topologies into a pruned space, thereby making the workflow-level topology search more efficient.

---

**Algorithm 1** MASS: Multi-Agent System Search

---

1: **Input**: Agentic modules in the search space $a_i \in \mathcal{A}$, workflow of agents $\mathcal{W}(a)$, prompt optimizer $\mathcal{O}$, evaluator $\mathcal{E}$, validation set $\mathcal{D}$, temperature $t$, number of candidates $N$, budget $B$.
2: **Output**: Optimized multi-agent system $\mathcal{W}^*$.
3: [**1PO**: *Block*-level **Prompt** Optimization]
4: Prompt optimization for the initial agent: $a_0^* \leftarrow \mathcal{O}_{\mathcal{D}}(a_0)$.
5: **for** $a_i$ in $\mathcal{A} \setminus \{a_0\}$ **do**
6:     Local prompt optimization for each building block in the design space: $a_i^* \leftarrow \mathcal{O}_{\mathcal{D}}(a_i | a_0^*)$.
7:     Obtain incremental Influence: $I_{a_i} \leftarrow \mathcal{E}(a_i^*)/\mathcal{E}(a_0^*)$.
8: **end for**
9: [**2TO**: Workflow **Topology** Optimization]
10: Obtain the selection probability $p_a \leftarrow \mathrm{Softmax}(I_a, t)$.
11: **while** $n < N$ **do**
12:     Search space pruning: $\mathcal{A}_p = \{a_i\}$ for $a_i$ in $\mathcal{A}$ if $u < p_{a_i}$, where $u \sim \mathrm{Uniform}(0, 1)$.
13:     Reject sampling for agentic configurations in budget: $a \sim \mathcal{A}_p$ s.t. $\mathcal{N}(a) < B$.
14:     Build the workflow $\mathcal{W}_c \leftarrow (a_i^*(\cdot), a_{i+1}^*(\cdot), \ldots)$ in a rule-base order with optimized prompts.
15:     Evaluate and record the score $\mathcal{E}_{\mathcal{D}}(\mathcal{W}_c)$.
16: **end while**
17: Obtain the best-performing workflow $\mathcal{W}_c^* \leftarrow \arg\max_{c \in \mathcal{C}} \mathcal{E}_{\mathcal{D}}(\mathcal{W}_c)$.
18: [**3PO**: *Workflow*-level **Prompt** Optimization]
19: Workflow-level prompt optimization for the best-performing topology: $\mathcal{W}^* \leftarrow \mathcal{O}_{\mathcal{D}}(\mathcal{W}_c^*)$.
20: **Return** optimized multi-agent system $\mathcal{W}^*$.

---

Here, we propose to measure the incremental influence $I_{a_i} = \mathcal{E}(a_i^*)/\mathcal{E}(a_0^*)$ that quantifies the relative gain for integrating the search dimension $a_i$ over the initial agent $a_0$. Following the intuition that influential dimension comes with higher selection probability, we reject the corresponding topology search dimension $a_i$ if $u > p_{a_i}$ to form a pruned search space, given $u \sim \mathrm{Uniform}(0, 1)$ and $p_a = \mathrm{Softmax}(I_a, t)$, where $t$ is the temperature for the softmax function. Therefore, the workflow can be randomly sampled from a pruned configuration space within a maximum budget, such that $a \sim \mathcal{A}_p$ s.t. $\mathcal{N}(a) < B$, where $\mathcal{N}(a)$ caps the overall number of agents and $B$ is the maximum budget; For instance, given $\mathcal{A} = \{a_i\}$ as the original configuration space with the topology building block search dimension $a_i$ parameterized by its property $N_{a_i}$ as defined in Sec. 2.2, each search dimension $a_i$ will be weighted by the influence of that dimension $p_{a_i}$, and rejected to form $\mathcal{A}_p$ if $u \sim \mathrm{Uniform}(0, 1) > p_{a_i}$. Followed by sampling valid configurations from $\mathcal{A}_p$, the workflow $\mathcal{W}(a) = (a_i, a_i + 1, \ldots)$ is then constructed in a predefined rule to arrange the flow of agents, where the rule-based flow removes redundancy in various orders in design and reduce the optimization complexity. The rule follows a simple predefined sequence that aligns with the practice of agent designs, such that [`summarize`, `reflect`, `debate`, `aggregate`]. We refer to App. §C for the detailed construction rule, MAS visualizations, and search space per task.

**3) Workflow-level prompt optimization.** As a final step, we treat the entire MAS design as an integrated entity and conduct a round of joint prompt optimization over all agents simultaneously, conditioned on the best topology discovered in Stage (2), $\mathcal{W}^* = \mathcal{O}_{\mathcal{D}}(\mathcal{W}_c^*)$. Although prompts were optimized at the individual level in Stage (1), this stage acts as an adaptation or fine-tuning process, ensuring that prompts are tailored for orchestration within the MAS and that the interdependence between agents is optimized appropriately. Our experiments (Fig. 5 & 6) demonstrate that this stage often yields practical benefits with a converged performance as shown in Table 6.

## 4 EXPERIMENTS

**Models and evaluation data.** Aside from the common benchmarks used for automating MAS (Hu et al., 2024a; Zhang et al., 2024b), we conduct experiments on an extensive collection of tasks: 1) Hendryck's MATH (Hendrycks et al., 2021) and DROP (Dua et al., 2019) for reasoning; HotpotQA (Yang et al., 2018), MuSiQue (Trivedi et al., 2022), 2WikiMultiHopQA (Ho et al., 2020) from LongBench (Bai et al., 2024) for long-context understanding; 3) MBPP (Austin et al., 2021), HumanEval (Chen et al., 2021), and LiveCodeBench (LCB) 'test output prediction' (Jain et al., 2024) for coding. We refer to App. §C & §E for details on data splits and prompt templates. We conduct all main experiments primarily on two Gemini 1.5 model sizes (Reid et al., 2024) (`gemini-1.5-{pro,flash}-002`) and further validate key findings

Table 1: Results on the evaluation set with Gemini 1.5 Pro and Gemini 1.5 Flash. We report the mean and standard deviation for all results with 3 runs of evaluations. We report the accuracy (%) for MATH and the test-output-prediction subtask of LiveCodeBench (LCB), F1 score for DROP, HotpotQA, MuSiQue, and 2WikiMQA, and pass@1 for MBPP and HumanEval. We note that the meta-prompt of AFlow[*] only works properly with Claude 3.5 Sonnet. Therefore, we reproduce AFlow with Gemini 1.5 Pro as the executor and Claude 3.5 Sonnet as the optimizer, where [*] indicates the results are only for reference. The inference cost is controlled comparably as shown in Table 7.

| Gemini-1.5-pro-002 | | | | | | | | | |
|---|---|---|---|---|---|---|---|---|---|
| **Task** | **Reasoning** | | **Multi-hop Long-context** | | | | **Coding** | | |
| Method | MATH | DROP | HotpotQA | MuSiQue | 2WikiMQA | MBPP | HumanEval | LCB | Avg. |
| CoT | $71.67_{3.30}$ | $70.59_{1.67}$ | $57.43_{0.52}$ | $37.81_{1.43}$ | $63.39_{1.12}$ | $68.33_{0.47}$ | $86.67_{0.94}$ | $66.33_{0.62}$ | 65.28 |
| Self-Consistency | $77.33_{1.25}$ | $74.06_{0.90}$ | $58.60_{2.19}$ | $41.81_{1.00}$ | $67.79_{1.19}$ | $69.50_{0.71}$ | $86.00_{0.82}$ | $70.33_{0.94}$ | 68.18 |
| Self-Refine | $79.67_{2.36}$ | $71.03_{1.31}$ | $60.62_{3.33}$ | $42.15_{1.34}$ | $66.74_{2.43}$ | $63.67_{0.24}$ | $84.00_{1.63}$ | $67.33_{1.31}$ | 66.90 |
| Multi-Agent Debate | $78.67_{0.94}$ | $71.78_{0.71}$ | $64.87_{0.23}$ | $46.00_{0.80}$ | $71.78_{0.63}$ | $68.67_{0.85}$ | $86.67_{1.25}$ | $73.67_{1.65}$ | 70.26 |
| ADAS | $80.00_{0.82}$ | $72.96_{0.90}$ | $65.88_{1.29}$ | $41.95_{1.24}$ | $71.14_{0.66}$ | $73.00_{1.08}$ | $87.67_{1.70}$ | $65.17_{1.25}$ | 69.72 |
| AFlow[*] | $76.00_{0.82}$ | $88.92_{0.63}$ | $68.62_{0.47}$ | $32.05_{1.29}$ | $\mathbf{76.51}_{1.05}$ | - | $88.00_{0.00}$ | - | - |
| MASS (Ours) | $\mathbf{84.67}_{0.47}$ | $\mathbf{90.52}_{0.64}$ | $\mathbf{69.91}_{1.11}$ | $\mathbf{51.40}_{0.42}$ | $73.34_{0.67}$ | $\mathbf{86.50}_{0.41}$ | $\mathbf{91.67}_{0.47}$ | $\mathbf{82.33}_{0.85}$ | **78.79** |
| Gemini-1.5-flash-002 | | | | | | | | | |
| CoT | $66.67_{2.36}$ | $71.79_{0.69}$ | $57.82_{1.10}$ | $37.10_{1.35}$ | $63.40_{0.68}$ | $63.33_{1.25}$ | $75.67_{1.89}$ | $51.17_{0.24}$ | 60.87 |
| Self-Consistency | $69.33_{1.25}$ | $73.42_{0.19}$ | $60.19_{1.01}$ | $41.94_{0.93}$ | $67.98_{0.72}$ | $63.67_{0.62}$ | $77.67_{1.89}$ | $53.83_{1.18}$ | 63.50 |
| Self-Refine | $71.33_{0.94}$ | $73.71_{1.09}$ | $58.84_{3.04}$ | $41.21_{1.99}$ | $65.56_{1.57}$ | $63.33_{1.25}$ | $81.67_{1.89}$ | $52.00_{1.41}$ | 63.46 |
| Multi-Agent Debate | $71.67_{0.94}$ | $74.79_{0.87}$ | $64.17_{1.69}$ | $46.27_{1.33}$ | $72.19_{0.54}$ | $63.00_{0.71}$ | $79.67_{1.25}$ | $55.50_{0.41}$ | 65.91 |
| ADAS | $68.00_{1.41}$ | $75.95_{1.18}$ | $61.36_{2.89}$ | $\mathbf{48.81}_{1.03}$ | $66.90_{1.00}$ | $65.83_{0.24}$ | $80.67_{2.49}$ | $50.50_{1.63}$ | 64.75 |
| MASS (Ours) | $\mathbf{81.00}_{2.45}$ | $\mathbf{91.68}_{0.14}$ | $\mathbf{66.53}_{0.38}$ | $43.67_{1.21}$ | $\mathbf{76.69}_{0.50}$ | $\mathbf{78.00}_{0.82}$ | $\mathbf{84.67}_{0.47}$ | $\mathbf{72.17}_{0.85}$ | **74.30** |

on Claude 3.5 Sonnet (`claude-3-5-sonnet@20240620`) (Anthropic, 2024) and Mistral Nemo (`mistral-nemo-12b`) (Mistral, 2024) in App. §D.

**Baselines.** We consider the following baselines: 1) CoT (Kojima et al., 2022): direct chain-of-thought reasoning via zero-shot prompting; 2) CoT-SC (Wang et al., 2023) (@9 agents): with self-consistency to find the most consistent answers from diversified reasoning traces; 3) Self-Refine (Madaan et al., 2024; Shinn et al., 2024) (2 agents @5 rounds): reflective agents to verify and self-refine predictions; 4) Multi-Agent Debate (Du et al., 2024; Liang et al., 2024) (3 agents @3 rounds + 1 judger): with agents justifying answers and aggregating information from other agents in multi-round debate; 5) ADAS (Hu et al., 2024a): an automatic agent design framework, where an LLM-based meta-agent iteratively proposes new agents based on former evaluations; 6) AFlow (Zhang et al., 2024b): automatic workflow design via Monte-Carlo Tree Search over a set of pre-defined operators. We fairly compare all baselines with a comparable inference cost per query and a maximum number of agents of 10. We refer to Table 7 for the detailed token consumption and App. §C for all specifications.

**Setup.** MASS is plug-and-play with arbitrary prompt optimizers. We integrate MIPRO (Opsahl-Ong et al., 2024) for a joint instruction and exemplar optimization, and we ablate other prompt optimizers implemented with MASS in Table 9. We limit the number of bootstrapped demonstrations to 3 and instruction candidates to 10, per agent in 10 rounds. In topology optimization for all tasks, we search for 10 different topologies via rejection sampling. Along with topology optimization, each topology is evaluated on the validation set 3 times to stabilize the prediction. The optimized MAS is then reported on the held-out test set over three runs. We set model temperature $T$ at 0.7, maximum output tokens at 4096, and the $t$ in Softmax at 0.05 for sharpening the selection probability $p_a$ for each search dimension. We implement the same LLM backbone as both evaluator and optimizer.

**Main results.** We present the main results of MASS compared to the baselines on the evaluation set in Table 1. MASS yields substantial gains over common forms of multi-agent systems, (e.g. self-consistency, self-refine, and multi-agent debate), that scale up without optimizing prompts for agents in collaboration. MASS leads to high-performing MAS: 78.8% and 74.3% on average on Gemini 1.5 Pro and Flash, respectively, where we observe consistent improvements on Claude 3.5 Sonnet and Mistral Nemo as reported in Table 4 & 5. By comparing MASS with state-of-the-art automatic agent design baselines, ADAS and AFlow, we first notice that ADAS only brings subtle gains even by already conditioning its meta-agent generation based on the common forms of agents. The meta-agent keeps proposing complex topologies but without optimizing the prompt design. AFlow, on the other hand, demonstrates a competitive performance to MASS, especially on 2WikiMQA and HumanEval.

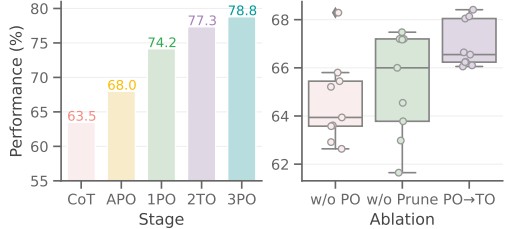

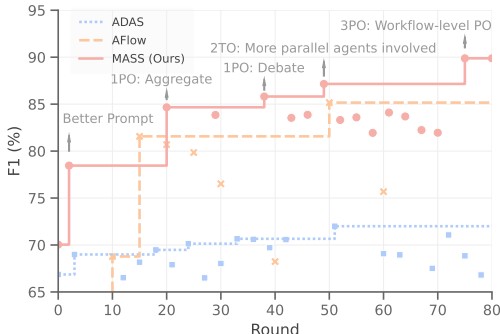

Figure 5: **Left**: average performance per optimization stage of MASS over 8 evaluation tasks on Gemini 1.5 Pro. We compare MASS with a single agent (CoT) starting point as the reference and an APO baseline that optimizes over the single agent by MIPROv2 (Opsahl-Ong et al., 2024). Refer to App. §D for the detailed ablation per task. **Right**: a comparative ablation study on topology optimization (2TO) without pruning and without the former stage of prompt optimization (1PO) evaluated on HotpotQA.

Figure 6: The optimization trajectories of MASS compared to agent design baselines per validation round on DROP. We note that, as a distinct advantage of MASS, the optimization within stages (1) & (2) of MASS can be completely parallelized, whereas ADAS and AFlow are iterative algorithms that have to wait to propose new agents until finishing earlier trajectories.

We attribute the performance of AFlow to: 1) its 'expansion' phase that generates new nodes based on an error log that contrasts the predictions with the ground truth, which provides implicit textual gradients (Pryzant et al., 2023) to reflect on any formatting errors in prompt design; 2) a more refined search space within a pre-defined set of operators. Though AFlow draws similar inspirations on the importance of search space design as MASS, it still lacks a phase of prompt optimization to *optimize* its pre-defined operators properly, resulting in under-performance for MAS search results at MATH and MuSiQue. Different from these baselines, the consistent improvements brought by MASS highlight the importance of searching in both prompt and topology design space.

**Ablating optimization stages.** To understand the incremental gain per MASS optimization stage, we provide a stage-by-stage ablation study in Fig. 5. We list the average performance of MASS from block-level to workflow-level optimization and compare it with a single agent APO baseline (MIPROv2), where the block-level optimization performance indicates the best-performing building block $a \in \mathcal{A}$ after APO. First, we notice that there is a large gain, 6% on average, between block-level optimization and single-agent APO, showing that MAS benefits substantially from having its agents optimized inside the building block, outperforming APO alone significantly. In addition, going from Stage (1) to (2), another 3% gain can be achieved by composing influential topologies while searching for the optimal configurations. Here, we provide an ablation on conducting Stage (2) without prompt optimization beforehand or without search space pruning. Fig. 5 (right) shows that both of them are critical for effective search space exploration. Lastly, MASS obtains further gains (∼2%) by conducting workflow-level prompt optimization on the best-found topology, which indicates that optimizing the prompts towards modeling the interdependence of agents is beneficial in MAS.

**Cost-effectiveness of MASS.** We conduct analysis on the cost-effectiveness of MASS. In particular, we visualize the optimization trajectory of MASS as shown in Fig. 6. MASS's trajectory demonstrates a steady trend of optimization that gradually improves the validation performance via interleaving the search towards better prompts and topologies. However, when it comes to automatic design baselines without explicit prompt optimization stages, AFlow is exposed to a larger variance in its optimization due to the nature of MCTS, whereas ADAS gets trapped in discovering over-complex topologies that appear to be less effective than the prompt design space. Overall, the optimization trajectory of MASS highlights the importance of optimizing in an effective design space, where interleaved optimization further resolves the complexity with more consecutive rewards within the same amount of training costs compared to baselines. Following Sec. 2.1, MASS also demonstrated advanced inference token-effectiveness and a comparable training cost, which we refer to Fig. 9 and Table 7.

**Best-found MAS architectures & Design principles.** We further inspect an example of optimized prompts and the trajectory of MASS in discovering more effective topologies in Fig. 7. The optimization starts from a zero-shot CoT agent, and soon MASS in Stage (1) identifies the high-performing topology in debate with its optimized prompt. However, as found in Stage (2), aggregating with more parallel agents actually outweighs the multi-agent debate. Workflow-level prompt optimization then

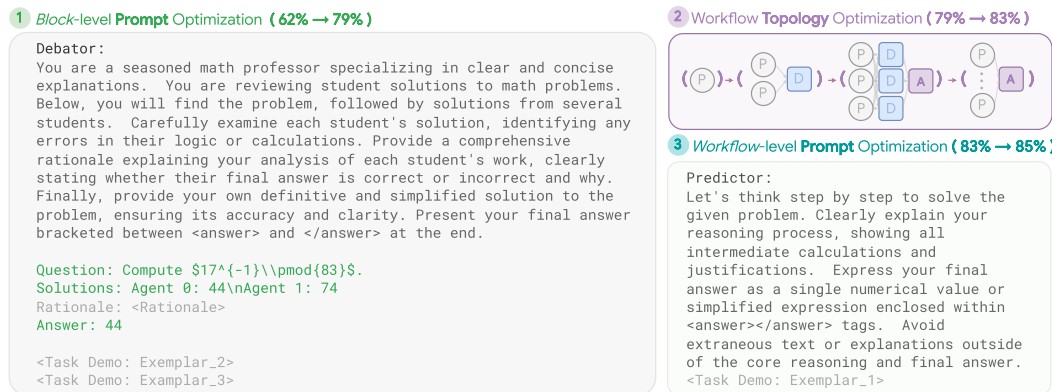

Figure 7: A demonstration of the optimization trajectory of MASS on MATH. In (1) block-level optimization: multi-agent debate serves as the best-performing topology. In (2) workflow **topology** optimization, aggregating with more parallel agents outweighs the performance of agents in debate. Lastly, (3) workflow-level optimization discovers the optimal prompt conditioned on the best topology.

leads to the best-performing predictor for aggregation. The overall optimization flow sheds light on our guidelines for building effective MAS: 1) Optimizing individual agents properly is important before composing them into an MAS; 2) A more effective MAS can be built by composing influential topologies; and 3) Modeling the interdependence between agents is beneficial, and can be achieved via workflow-level joint optimization.

## 5  RELATED WORK

**Forms of LLM-based agentic systems.** The simplest form of an LLM-based agentic system involves a single agent that can dynamically interact and respond to the environment (Yao et al., 2023). Recent advances endow agents with diverse roles and tools (Wu et al., 2023), orchestrating multiple agents to cooperate with each other (Chen et al., 2024b). Standard forms of agent cooperation (i.e., topology) often involve parallel and serial flows of information. The parallel form usually diversifies the exploration among many agents in parallel (Li et al., 2024a), and self-consistency (SC) (Wang et al., 2023) is a representative way for scaling agents in parallel. The serial form aims to advance the exploitation of a task via a chain of agents, where LLMs can serve as reflective agents to self-justify and refine former predictions (Madaan et al., 2024; Shinn et al., 2024). Later, the opinions from multiple agents can be summarized to retrieve the most consistent answer by an aggregation agent (Chen et al., 2024c; Lin et al., 2024). Moreover, multi-agent debate consists of a more complex flow of information (Chen et al., 2024a; Wang et al., 2024c; Zhang et al., 2024c), and recent research shows that debating can elicit more truthful predictions (Khan et al., 2024; Du et al., 2024). Recent agent topology extends beyond the above connections (Wang et al., 2024b; Qian et al., 2024), and MASS can automatically search the best topology among the aforementioned spaces.

**Automatic optimization for MAS.** Recent research starts automating agent design by interpreting agent functions as learnable policies (Zhang et al., 2024d;e) and synthesizing trajectories for agent fine-tuning (Qiao et al., 2024). Going further from a single agent, automatic multi-agent optimization faces a higher level of complexity, thereby requiring a more sophisticated design of the search space and algorithms. Among all recent advances in multi-agent optimization, the optimization space has spanned prompts (Khattab et al., 2024; Wang et al., 2024d), tools (Zhou et al., 2024c), workflows (Li et al., 2024c), and thinking strategies (Shang et al., 2024). Aligning closer to our topology search space, DyLAN (Liu et al., 2024b) dynamically activates the composition of agents, and Archon (Saad-Falcon et al., 2024) frames MAS as a hyperparameter optimization problem. Neither of them has taken the important prompt space into account, where we demonstrated the importance of prompt optimization in Sec. 3. In addition, GPTSwarm (Zhuge et al., 2024) optimizes the connections between agentic nodes using a policy gradient algorithm. State-of-the-art automatic agent design methods, ADAS (Hu et al., 2024a) and AFlow (Zhang et al., 2024b), also attempt to optimize agentic workflows with advanced search algorithms and LLM as optimizers. Concurrently with us, MaAS (Zhang et al., 2025) optimizes an agentic supernet for query-dependent MAS. However, we observe that the importance of prompt designs has been relatively under-studied in these prior works.

## 6 CONCLUSION

We approach designing effective MAS by first conducting a thorough analysis of the massive design space, revealing the crucial role of prompts, and identifying the redundancy in MAS search space design. Building on these findings, we introduce MASS, a novel multi-stage optimization framework that searches within a pruned design space, interleaving prompt and topology optimization to efficiently generate high-performing MAS. MASS is agnostic to customized prompt optimizers and topology design space. Our experiments demonstrate that MASS-optimized MAS significantly outperforms existing manual and automated approaches across an extensive set of tasks. Finally, based on the optimized systems discovered by MASS, we extract valuable design principles to guide the development of future MAS, and we expect future prospective prompt and topology designs integrated with MASS towards building more efficient and effective LLM-based agents.

## ACKNOWLEDGMENT

We thank Jinsung Yoon and all other colleagues from Google Cloud AI Research for their valuable feedback. The work has been supported by the UK Research and Innovation (UKRI) Frontier Research Grant EP/Y031350/1 (the UK government's funding guarantee for ERC Advanced Grants) awarded to Anna Korhonen at the University of Cambridge. The work has also been supported in part by a Royal Society University Research Fellowship (no 221137; 2022-) awarded to Ivan Vulić, and by the UK EPSRC grant EP/T02450X/1.

## REPRODUCIBILITY STATEMENT

We include detailed implementation details with hyperparameter settings in Sec. 4 and Appendix. C. The specification of datasets is listed in Table 2. We list the MASS search space in Table 3 and the multi-agent construction rules in Appendix. C.3. In addition, we provide the visualization of both the minimum building blocks and the optimized topology in Fig. 8. Furthermore, we provide the full prompt templates to each agent to fully reproduce the main experiments in Appendix. E.

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

## A  The Use of Large Language Models

Large language models (LLMs) were used as general-purpose assist tools in this work. Specifically, LLMs assisted in polishing the writing to improve clarity and readability.

## B  Limitations and future work

MASS is a multi-agent design meta-framework also orthogonal to prompt and topology optimizers. MASS has brought substantial improvements over a single agent design by searching in a customizable topology space. Though our proposed topology space has covered the vast majority of effective MAS designs, including serial, parallel, and mixture of connections, it is still likely that incorporating other topologies may further improve the final performance of MASS, which is complementary to the development of MASS. For instance, the debate topology proposed in MASS involves a fully-connected topology across agents. Recent work has been identifying the sparsity of agent communications (Li et al., 2024b; Zhang et al., 2024a), and pruning redundant communications may further enhance the overall efficiency of the strongest MASS-found design. Though the topology optimizer in MASS already traverses efficiently in the proposed topology space, incorporating more advanced search algorithms, such as the Bayes optimizer (Kandasamy et al., 2018; Ru et al., 2021), may further improve the sample efficiency of MASS when faces a more complex design space. Similarly, the sample efficiency of the prompt optimizer may be further enhanced by conditioning on textual feedback from error logs (Pryzant et al., 2023; Cheng et al., 2024), which we will endeavor to explore in future work. These prospective future topology and prompt design methods will further strengthen MASS, while MASS has been served as a generalizable guideline in identifying influential agentic design components and a systematic framework for scalable MAS optimization.

## C  Implementation details

### C.1  Datasets

In this work, we included the following dataset: 1) Hendryck's MATH (Hendrycks et al., 2021) consisting competition-level mathematics problems, and DROP (Dua et al., 2019) requires discrete and symbolic reasoning over paragraphs; 2) HotpotQA (Yang et al., 2018), MuSiQue (Trivedi et al., 2022), and 2WikiMultiHopQA (Ho et al., 2020) to evaluate long-context reasoning with agentic systems, which we report from standardized versions in LongBench (Bai et al., 2024); 3) MBPP (Austin et al., 2021), HumanEval (Chen et al., 2021), and LiveCodeBench (Jain et al., 2024) as well-established coding benchmarks. Regarding LiveCodeBench, we use the 'test output prediction' task as an agent cooperative task. In line with AFlow (Zhang et al., 2024b), we use the public test cases of MBPP and HumanEval for the executor to retrieve reliable external feedback signals. To save computation resources, we randomly sample a subset of the original validation and test splits to conduct all the experiments, where the specifications are reported in Table 2.

Table 2: The specification of evaluation tasks: dataset split, topology search space, and the MASS-optimized MAS (on Gemini 1.5 Pro).

| Task | Type | \|Val\| | \|Test\| | Topology Search Space | MASS |
|---|---|---|---|---|---|
| MATH | Mathematical Reasoning | 60 | 100 | {Aggregate, Reflect, Debate} | {9, 0, 0} |
| DROP | Discrete Reasoning | 60 | 200 | {Aggregate, Reflect, Debate} | {5, 0, 0} |
| HotpotQA | Long-context Understanding | 50 | 100 | {Summarize, Aggregate, Reflect, Debate} | {0, 5, 0, 1} |
| MuSiQue | Long-context Understanding | 50 | 100 | {Summarize, Aggregate, Reflect, Debate} | {0, 3, 0, 2} |
| 2WikiMQA | Long-context Understanding | 50 | 100 | {Summarize, Aggregate, Reflect, Debate} | {0, 3, 0, 1} |
| MBPP | Coding | 60 | 200 | {Aggregate, Reflect, Debate, Executor} | {1, 4, 0, 1} |
| HumanEval | Coding | 50 | 100 | {Aggregate, Reflect, Debate, Executor} | {1, 3, 0, 1} |
| LiveCodeBench | Coding: test output prediction | 100 | 200 | {Aggregate, Reflect, Debate, Executor} | {3, 1, 1, 1} |

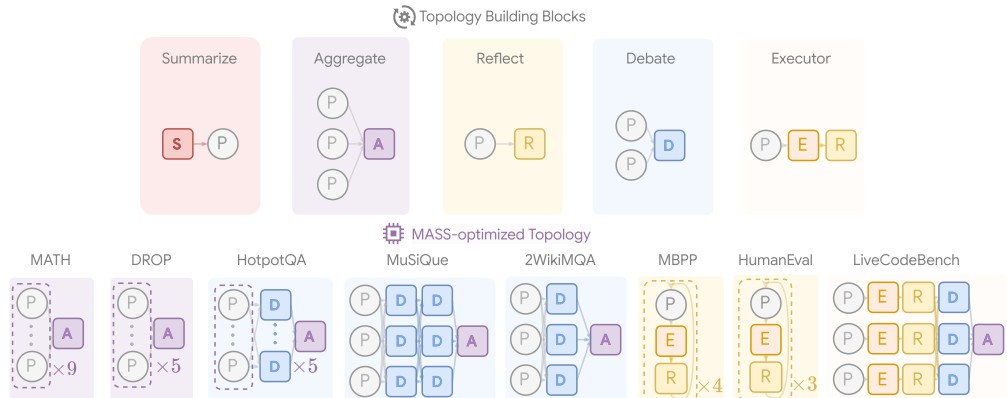

Figure 8: Visualization of the topology building blocks and best MASS-discovered topologies from Gemini 1.5 Pro.

Table 3: The search dimension for each topology. The minimum topology defines the building block that MASS Stage (1) optimized. We refer the definition of search space to Sec.2.2.

| Topology | Search Space | Minimum Topology Building Block | Definition |
|---|---|---|---|
| Summarize | $N_s \in \{0, 1, 2, 3, 4\}$ | 1 Summarizer + 1 Predictor | Rounds of summarization |
| Aggregate | $N_a \in \{1, 3, 5, 7, 9\}$ | 3 Predictor + 1 Aggregator | Number of parallel agents |
| Reflect | $N_r \in \{0, 1, 2, 3, 4\}$ | 1 Predictor + 1 Reflector | Rounds of self-reflection |
| Debate | $N_d \in \{0, 1, 2, 3, 4\}$ | 2 Predictor + 1 Debator | Rounds of debating |
| Execute | $N_t \in \{0, 1\}$ | 1 Predictor + 1 Executor + 1 Reflector} | Use of code execution |

## C.2 BASELINES

In this section, we report the specifications of all our baselines. We note that for the baselines: CoT, SC, Self-Refine, and Multi-Agent Debate, we follow the prompts given in ADAS (Hu et al., 2024a).

1) Chain-of-Thought (CoT) (Kojima et al., 2022). Direct chain-of-thought reasoning via zero-shot prompting: "Please think step by step and then solve the task."

2) Self-Consistency (SC) (Wang et al., 2023). In self-consistency, we generate diverse chain-of-thought reasoning traces with a temperature of 0.8, followed by a rule-based majority vote that collects the most consistent answer. In Table 1, we report SC@9 to provide a fair comparison across baselines.

3) Self-Refine (Madaan et al., 2024): This baseline consists of one predictor that constantly takes feedback and a self-reflector that provides criticism. It involves a stop criterion whenever the self-reflector outputs "correct" in its prediction. We set the maximum number of rounds of reflections to 5, such that the worst case will involve 11 $(1 + 2 \times 5)$ calls.

4) Multi-Agent Debate (Du et al., 2024; Liang et al., 2024). In this baseline, it involves 3 agents that conduct reasoning and debating for 3 rounds. The opinions along the rounds of debating are finally judged by an aggregator that makes the final prediction. Hence, it contains 10 $(3 \times 3 + 1)$ agents.

5) Automated Design of Agentic Systems (ADAS) (Hu et al., 2024a). Consistent with our main experimental setups. We use Gemini 1.5 as both LLM optimizer and evaluator for reproducing all ADAS results. The generation of ADAS is conditioned on former evaluations of baselines, including CoT, SC, Self-Refine, and Multi-Agent Debate. We report ADAS with 30 rounds of search, and each round is evaluated on the validation set 3 times to stablize the prediction.

6) AFlow (Zhang et al., 2024b). Automatic workflow design via Monte-Carlo Tree Search over a set of pre-defined operators. Similar to ADAS, AFlow also relies on an LLM optimizer to generate new nodes and topologies expressed in codes. However, we find the meta-prompt of AFlow does not generalize to other LLM backbones. Consequently, we report AFlow with its original LLM optimizer by Claude 3.5 Sonnet, and reproduce experiments with Gemini 1.5 Pro as the LLM executor.

Therefore, the comparison is not completely fair, and we treat the results from AFlow as a good reference. We note that the '-' in Table 1 refers to out-of-time errors, where the LLM executor has been trapped in executing accidental scripts with infinite loops. We still endeavored to report most results from AFlow as shown in Table 1 & Fig. 6 with the default experimental setup from AFlow: 20 rounds, 5 runs of validation per round, and k at 3.

### C.3 MASS DETAILS AND CONSTRUCTION RULES

In this section, we provide additional details for MASS. The topology search space for each task is defined in Table 2. In addition, for Stage (1) block-level prompt optimization, the specification of the building block is defined in Table 3. MASS follows a simple and predefined rule for constructing different topologies in sequence. Throughout our study, we find that the impact of different agentic configurations outweighs the ordering substantially. Consequently, our rule follows the practice of agent designs by constructing topology in a fixed order of `[summarize, reflect, debate, aggregate]`. When multiple search dimensions are kept active, `aggregate` basically controls the number of chain-of-agents in parallel, where the length of the chain is defined via the pre-defined order. We provide the visualization of both the minimum building blocks and the optimized topology in Fig. 8. We refer the reader to App. §E & §F for the prompt templates we used to define each type of agent and the best prompts discovered.

## D  ADDITIONAL EXPERIMENTS

### D.1  GENERALIZATION ACROSS LLM BACKBONES

Table 4: Results on the evaluation set with Claude 3.5 Sonnet. We keep the same experimental setup as Table 1. Since Claude 3.5 Sonnet does not support the same context window as Gemini, we report the standard HotpotQA instead of the LongBench. As we transfer the prompt template for each agent from Gemini to Claude, it is noticeable that the basic topology on some tasks may result in severe degradation of performance, and MASS successfully recovers the performance and brings significant improvements over the initial agent.

| Claude-3.5-Sonnet | | | | | | | |
|---|---|---|---|---|---|---|---|
| **Task** | **Reasoning** | | **Multi-hop** | | **Coding** | | |
| Method | MATH | DROP | HotpotQA | MBPP | HumanEval | LCB | Avg. |
| CoT | $57.33_{0.94}$ | $55.52_{0.42}$ | $23.56_{1.52}$ | $67.50_{1.47}$ | $88.67_{1.70}$ | $72.67_{2.39}$ | 60.21 |
| Self-Consistency | $61.67_{1.89}$ | $57.86_{0.45}$ | $25.69_{0.44}$ | $\mathbf{69.17}_{0.62}$ | $90.00_{0.82}$ | $72.67_{2.39}$ | 62.84 |
| Self-Refine | $57.00_{1.63}$ | $56.26_{0.56}$ | $23.57_{2.56}$ | $68.00_{0.82}$ | $87.00_{1.41}$ | $49.33_{1.65}$ | 56.86 |
| Multi-Agent Debate | $45.00_{3.74}$ | $26.62_{0.11}$ | $31.41_{3.30}$ | $00.00_{0.00}$ | $84.33_{3.30}$ | $72.82_{1.84}$ | 43.36 |
| MASS | $\mathbf{63.00}_{0.00}$ | $\mathbf{68.93}_{0.38}$ | $\mathbf{66.98}_{0.99}$ | $68.83_{0.62}$ | $\mathbf{93.00}_{0.82}$ | $\mathbf{73.73}_{1.43}$ | $\mathbf{72.43}$ |

Table 5: Results on the evaluation set with the open-source model, Mistral-Nemo-12B. We keep the same experimental setup as Table 4 and evaluate a subset of representative coding tasks to save resources. MASS demonstrate consistent improvements over the baselines on Mistral Nemo.

| Mistral-Nemo-12B | | | | | |
|---|---|---|---|---|---|
| **Task** | **Reasoning** | | **Multi-hop** | **Coding** | |
| Method | MATH | DROP | HotpotQA | MBPP | Avg. |
| CoT | 13.3 | 49.0 | 55.9 | 43.5 | 40.4 |
| Self-Consistency | 22.0 | 57.6 | 58.9 | 46.5 | 46.3 |
| Self-Refine | 14.3 | 48.6 | 52.5 | 48.0 | 40.9 |
| Multi-Agent Debate | 26.0 | 61.4 | 59.5 | 40.5 | 46.9 |
| MASS | $\mathbf{43.7}$ | $\mathbf{68.4}$ | $\mathbf{62.6}$ | $\mathbf{48.7}$ | $\mathbf{55.9}$ |

### D.2  DISCUSSION ON MASS-FOUND TOPOLOGIES

In Fig. 8, we find that the optimal MASS-found topologies indicate certain patterns per task family, and there are topologies that demonstrate clear advantages over other topologies in particular tasks.

Table 6: The detailed ablation results per optimization stage of MASS. Practical gains can be obtained by further conducting workflow-level prompt optimization (3PO) on the best-found topology.

| Task | **Reasoning** | | **Multi-hop Long-context** | | | | **Coding** | | | |
|---|---|---|---|---|---|---|---|---|---|---|
| Method | MATH | DROP | HotpotQA | MuSiQue | 2WikiMQA | MBPP | HumanEval | LCB | Avg. |
| | | | | | | | | | | |
| Base Agent | $62.33_{0.94}$ | $71.65_{0.61}$ | $56.96_{1.26}$ | $43.32_{0.13}$ | $49.20_{0.61}$ | $68.83_{0.85}$ | $89.33_{1.70}$ | $66.33_{2.09}$ | 63.54 |
| + APO | $79.33_{1.89}$ | $77.51_{0.38}$ | $59.72_{0.00}$ | $43.97_{0.00}$ | $61.49_{0.24}$ | $67.00_{1.08}$ | $86.33_{1.25}$ | $68.50_{1.22}$ | 67.44 |
| + 1PO | $80.00_{0.00}$ | $86.45_{0.90}$ | $62.52_{1.86}$ | $48.86_{0.61}$ | $67.40_{0.58}$ | $80.33_{1.25}$ | $91.67_{1.25}$ | $76.00_{0.00}$ | 74.56 |
| + 2TO | $83.00_{1.63}$ | $86.75_{1.32}$ | $65.22_{1.34}$ | $52.61_{0.52}$ | $72.82_{0.86}$ | $85.00_{1.08}$ | $92.00_{0.82}$ | $81.33_{0.00}$ | 77.55 |
| + 3PO | $84.67_{0.47}$ | $90.52_{0.64}$ | $69.91_{1.11}$ | $51.40_{0.42}$ | $73.34_{0.67}$ | $86.50_{0.41}$ | $91.67_{0.47}$ | $82.33_{0.85}$ | 78.40 |

Gemini-1.5-pro-002

Table 7: The training and inference cost for running MASS and baselines, where we show the training cost and the actual run-time of MASS is comparable to the training cost of auto-agent baselines. We note that the performance of self-consistency, self-refine, and multi-agent debate is already saturated, and further scaling the inference cost of these baselines only brings marginal gains, whereas the MASS-found MAS outperforms the baseline substantially at a comparable inference token cost.

| | **Training** | | | **Inference** (per query) | | | |
|---|---|---|---|---|---|---|---|
| Method | Input Token | Output Token | Cost ($) | Input Token | Output Token | Cost ($) | Acc (%) |
| Self-Consistency | - | - | - | 1538 | 3013 | 0.0010 | 69.3 |
| Self-Refine | - | - | - | 2051 | 850 | 0.0004 | 71.3 |
| Multi-Agent Debate | - | - | - | 6536 | 2483 | 0.0012 | 71.7 |
| AFlow | 11M | 8 M | 3.89 | 2523 | 1481 | 0.0006 | 64.3 |
| ADAS | 23 M | 13 M | 5.61 | 7850 | 3335 | 0.0016 | 72.7 |
| MASS | 24 M | 11 M | 5.09 | 6645 | 3263 | 0.0014 | 81.0 |

By inspecting Fig. 8, we notice that the "debating" topology brings significant gains to all multi-hop tasks that require factual knowledge: HotpotQA, MuSiQue, and 2WikiMQA, which is aligned with previous multi-agent debate work (Khan et al., 2024) that argues debating will elicit more truthful answers. Reasoning tasks: MATH and DROP benefit from more exploration, where SC becomes more effective. Lastly, the coding tasks share a common pattern of reflection with tool-using. However, even the best configuration in the same task family still shows differentiations, indicating the necessity of automatic optimization. Therefore, no matter the underlying complexity of the task-dependent topology, the unique advantage of MASS is being able to identify the most influential topology automatically for any customized search space.

## D.3 COST ANALYSIS

To estimate the computation cost, the prompt optimizer evaluates the MAS with $K$ rounds of evaluation, where each round consists of $M$ prompt candidates, and each inference requires $\mathcal{N}(a)$ API calls by the number of agents in the MAS. Hence, 1PO scales linearly with the number of building blocks $J$, and its cost can be computed as $C(1\text{PO}) = \sum_j^J \mathcal{N}(a_j) \times M \times K$. The topology optimization evaluates $N$ topology candidates. Hence, $C(2\text{TO}) = \sum_n^N \mathcal{N}(\mathcal{W}_n)$. Similar to 1PO, 3PO conducts PO on the best-found workflow $\mathcal{W}^*$, such that $C(3\text{PO}) = \mathcal{N}(\mathcal{W}^*) \times M \times K$, and the cost of 3PO scales linearly with the number of agents in the network. The final computation cost is therefore a summation of the cost at individual stages.

In Table 7, we report the detailed token cost for both training MASS and the inference cost per query with reference costs from baselines, where we show that the training cost of MASS is comparable to the training cost of auto-agent baselines. It is worth noting that the performance of training-free baselines (e.g., self-consistency@9) is already saturated, and further scaling the inference cost of these baselines only brings marginal gains, whereas the MASS-found MAS outperforms baselines substantially. In addition, we record the Pareto-front of MASS optimized designs in Fig. 9. Though the primary objective of this work is a single-objective optimization that targets to maximize the task performance within the same training budget, we show that MASS can generate a Pareto-front of

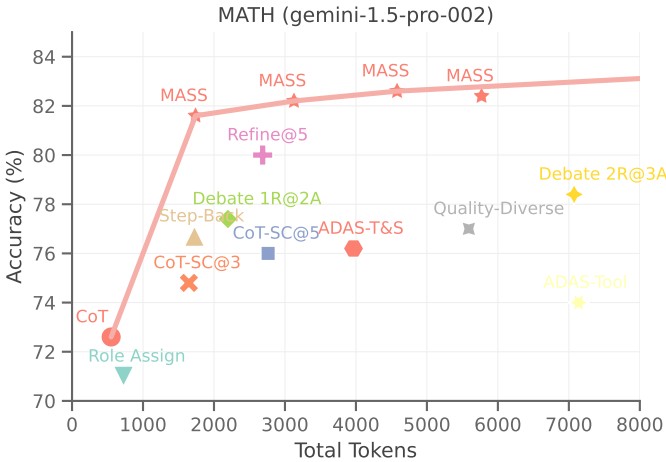

Figure 9: The Pareto-front of MASS-optimized designs compared to multi-agent baselines. Total tokens include both inference input tokens and output tokens. Additional multi-agent baselines from ADAS (Hu et al., 2024a) and the two best-found ADAS designs are included.

Table 8: Comparison of MASS with graph optimization baselines. We reproduce GPTSwarm with (Zhang et al., 2024a), and (Pro) & (Flash) indicate optimization results from Gemini 1.5 Pro and Gemini 1.5 Flash, respectively.

| Method | MATH (%) | HumanEval (%) | Average (%) |
|---|---|---|---|
| GPTSwarm (Pro) | 76.0 | 85.0 | 80.5 |
| MASS (Pro) | 84.7 | 91.7 | 88.2 |
| GPTSwarm (Flash) | 61.0 | 73.0 | 67.0 |
| MASS (Flash) | 81.0 | 84.7 | 82.9 |

optimized MAS designs with stronger token-effectiveness than baselines, and the more cost-efficient workflows can be selected from the pivotal points in the MASS Pareto-front.

## D.4 GRAPH OPTIMIZATION BASELINE

We further compare MASS against a graph optimization baseline, GPTSwarm (Zhuge et al., 2024), on the overlapped set of tasks from the original work. We observe that the graph optimization is more effective in improving the inference efficiency from a fully-connected graph to a sparse graph rather than enhancing the task performance, whereas the prompt optimization component of MASS particularly led to more significant contributions. Overall, MASS brought a substantial gain (8% and 6% over MATH & HumanEval, respectively) in representative reasoning and coding tasks compared to graph optimization methods.

## D.5 ABLATION ON PROMPT OPTIMIZERS

MASS is a plug-and-play framework with arbitrary prompt optimizers. We integrate MIPRO (Opsahl-Ong et al., 2024) as a representative prompt optimizer due to the importance of simultaneous instruction and exemplar optimization, which has been justified in both (Wan et al., 2024; Opsahl-Ong et al., 2024) that show superior performance over OPRO-style (Yang et al., 2024) instruction-only optimization methods. It is also worth noting that the MASS framework itself is agnostic to the prompt optimizer, and thus any prospective better methods can only enhance the overall performance of MASS. In Table 9, we additionally provide an ablation of common prompt optimizers, APE (Zhou et al., 2023b) & DSPy (Khattab et al., 2024), and we show MASS with exemplar optimization (+DSPy) also led to significant gains. We consider extending the existing PO to feedback-based optimizers (e.g., ProTeGi (Pryzant et al., 2023) or TextGrad (Yuksekgonul et al., 2025)) that may come with better sample efficiency as a desirable future work.

Table 9: Ablation of MASS with different prompt optimizers on Gemini 1.5 Flash.

| Method | CoT | MASS (APE) | MASS (DSPy) | MASS (MIPRO) |
|---|---|---|---|---|
| MATH (%) | 66.7 | 73.3 | 78.2 | 81.0 |

## E PROMPT TEMPLATE

We provide all the prompt templates we used for defining the MASS search space. We use <> to enclose texts that have been skipped for presentation purposes. We follow the DSPy (Khattab et al., 2024) in constructing these agentic templates.

The general template for instruction, exemplar, and input/output fields:

```
<Instruction>

---

Follow the following format.

Input: ${Input}
...
Output: ${output}

---

<example_1>

---

Input: <Input>
...
Output: <output>
```

MATH:

```
Predictor:

Let's think step by step.
---
Question: ${question}
Reasoning: Let's think step by step in order to ${produce the answer}. We ...
Answer: ${answer}

------------
Reflector:

Please review the answer above and criticize on where might be wrong. If you are absolutely
    sure it is correct, output 'True' in 'correctness'.

---
Question: ${question}
Text: ${text}
Reasoning: Let's think step by step in order to ${produce the correctness}. We ...
Feedback: ${feedback}
Correctness: True/False indicating if answer is correct given the question.

------------
Refiner:

Given previous attempts and feedback, carefully consider where you could go wrong in your
    latest attempt. Using insights from previous attempts, try to solve the task better. Show
     your final answer bracketed between <answer> and </answer> at the end.

---
Question: ${question}
Previous answer: ${previous_answer}
Reflection: ${reflection}
Correctness: ${correctness}
Thinking: ${thinking}
Answer: ${answer}

------------
Debator:
```

```
These are the solutions to the question from other agents. Examine the solutions from other
    agents in your rationale, finish by giving an updated answer. Show your final answer
    bracketed between <answer> and </answer> at the end.

---
Question: ${question}
Solutions: the solutions to the question from other agents
Reasoning: Let's think step by step in order to ${Examine the solutions from other agents}. We
    ...
Answer: The updated answer for the question. Do not repeat Answer:
```

## DROP:

```
Predictor:

Please think step by step and then solve the task. # Your Task:
Please answer the following question based on the given context.
---
Question: ${question}
Context: ${context}
Thinking: ${thinking}
Answer: Directly answer the question. Keep it very concise.

------------
Reflector:

Verify that the answer is based on the provided context. Give your reflection in the rationale
    .

---
Question: ${question}
Context: ${context}
Text: ${text}
Reasoning: Let's think step by step in order to ${produce the correctness}. We ...
Correctness: True/False indicating if answer is correct given the observations and question.

------------
Refiner:

Please think step by step and then solve the task. # Your Task:
Based on the reflection, correctness of the previous answer, and the context again, give an
    updated answer.

---
Question: ${question}
Context: ${context}
Previous answer: ${previous_answer}
Reflection: ${reflection}
Correctness: ${correctness}
Thinking: ${thinking}
Answer: Directly answer the question. Keep it very concise.

------------

Debator:

These are the solutions to the question from other agents. Based on the context, examine the
    solutions from other agents in your rationale, finish by giving an updated answer.

---
Question: ${question}
Context: ${context}
Solutions: the solutions to the question from other agents
Reasoning: Let's think step by step in order to ${Examine the solutions from other agents}. We
    ...
Answer: The updated answer for the question. Do not repeat Answer:
```

## HotpotQA, MuSiQue, and 2WikiMQA:

```
Predictor:

Answer the question with information based on the context. Only return the answer as your
    output.
---
Question: ${question}
Context: ${context}
Answer: Only give me the answer. Do not output any other words.

------------
Summarizer:
```

```
Based on the question, retrieve relevant information from context that is ONLY helpful in
    answering the question. Include all key information. Do not repeat context.
---
Question: ${question}
Context: ${context}
Summary: Only generate the summary. Start with Summary:

------------
Reflector:

Verify that the answer is based on the provided context.

---
Question: ${question}
Context: ${context}
Text: ${text}
Reasoning: Let's think step by step in order to ${produce the correctness}. We ...
Correctness: True/False indicating if answer is correct given the observations and question.

------------

Debator:

These are the solutions to the question from other agents. Based on the context, examine the
    solutions from other agents in your rationale, finish by giving an updated answer.

---
Question: ${question}
Context: ${context}
Solutions: the solutions to the question from other agents
Reasoning: Let's think step by step in order to ${Examine the solutions from other agents}. We
        ...
Answer: The updated answer for the question. Do not repeat Answer:
```

## MBPP:

```
Predictor:

Let's think step by step. Provide a complete and correct code implementation in python.
---
Question: ${question}
Thinking: ${thinking}
Answer: Only the code implementation. Do not include example usage or explainations.

------------
Reflector:

Please determine the correctness of the solution in passing all test cases. If it fails, based
        on the error message and trackback, think step by step, carefully propose an updated
    solution in the answer output with a correct code implementation in python.

---
Question: ${question}
Previous solution: ${previous_solution}
Traceback: It contains the test cases, execution results, and ground truth. If there is an
    error, the relevant traceback is given.
Correctness: 'True/False' based on the correctness of executive feedback. If there is an error
        message, output 'False'
Thinking: ${thinking}
Answer: ${answer}

------------

Debator:

These are the solutions to the question from other agents. Examine the solutions from other
    agents in your rationale, finish by giving an updated answer. Let's think step by step.
    Provide a complete and correct code implementation in python.

---
Question: ${question}
Solutions: the solutions to the question from other agents
Reasoning: Let's think step by step in order to ${Examine the solutions from other agents}. We
        ...
Answer: ${answer}
```

## HumanEval:

```
Predictor:

Let's think step by step. Provide a complete and correct code implementation in python.
---
```

```
Question: ${question}
Thinking: ${thinking}
Answer: ${answer}

------------
Reflector:

Please determine the correctness of the solution in passing all test cases. If it fails, based
     on the error message and trackback, think step by step, carefully propose an updated
     solution in the answer output with a correct code implementation in python.

---
Question: ${question}
Previous solution: ${previous_solution}
Traceback: ${traceback}
Thinking: ${thinking}
Answer: ${answer}

------------

Debator:

These are the solutions to the question from other agents. Examine the solutions from other
     agents in your rationale, finish by giving an updated answer. Let's think step by step.
     Provide a complete and correct code implementation in python.

---
Question: ${question}
Solutions: the solutions to the question from other agents
Reasoning: Let's think step by step in order to ${Examine the solutions from other agents}. We
     ...
Answer: ${answer}
```

## LiveCodeBench:

```
Predictor:

You are a helpful programming assistant and an expert Python programmer. The user has written
     a input for the testcase. Think step by step. You will generate the code based on the
     problem requirepement. You will calculate the output of the testcase and write the whole
     assertion statement in the markdown code block with the correct output.
---
Question: ${question}
Thinking: ${thinking}
Code: ${code}
Answer: complete the testcase with assertion.

------------
Reflector:

If there is an executive output in the traceback, parse the output into an assertion in the
     answer given the executive output.

---
Question: ${question}
Previous solution: ${previous_solution}
Traceback: It contains the test cases, execution results, and ground truth. If there is an
     error, the relevant traceback is given.
Correctness: 'True/False' based on the correctness of executive feedback. If there is an error
      message, output 'False'
Thinking: ${thinking}
Answer: ${answer}

------------

Debator:

These are the solutions to the question from other agents. Examine the solutions from other
     agents in your rationale, finish by giving an updated answer.

---
Question: ${question}
Solutions: the solutions to the question from other agents
Reasoning: Let's think step by step in order to ${Examine the solutions from other agents}. We
     ...
Answer: assert ${function(input)} == {executive_output}
```

## F  BEST PROMPTS DISCOVERED

In this section, we show a few typical prompts that have been discovered by MASS. Similarly, we indicate <> that skips some comments.

MATH:

```
Predictor:

Let's think step by step to solve the given problem. Clearly explain your reasoning process,
    showing all intermediate calculations and justifications.  Express your final answer as a
     single numerical value or simplified expression enclosed within <answer></answer> tags.
     Avoid extraneous text or explanations outside of the core reasoning and final answer.

---

Follow the following format.

Question: ${question}
Reasoning: Let's think step by step in order to ${produce the answer}. We ...
Answer: ${answer}

---

Question: What is the average of the two smallest positive integer solutions to the congruence
     $$14u \\equiv 46 \\pmod{100}~?$$
Reasoning: Let's think step by step in order to find the average of the two smallest positive
    integer solutions to the congruence $14u \\equiv 46 \\pmod{100}$. We are looking for
    solutions to $14u \\equiv 46 \\pmod{100}$. We can divide the congruence by 2 to get $7u
    \\equiv 23 \\pmod{50}$. We want to find the inverse of 7 modulo 50. We use the Euclidean
    Algorithm: \\begin{align*} 50 &= 7 \\cdot 7 + 1 \\\\ 1 &= 50 - 7 \\cdot 7 \\end{align*}
    Thus, $7(-7) \\equiv 1 \\pmod{50}$, so $7^{-1} \\equiv -7 \\equiv 43 \\pmod{50}$.
    Multiplying $7u \\equiv 23 \\pmod{50}$ by 43 gives $u \\equiv 23 \\cdot 43 \\pmod{50}$.
    $23 \\cdot 43 = 989 = 50 \\cdot 19 + 39$, so $23 \\cdot 43 \\equiv 39 \\pmod{50}$.
    Therefore, $u \\equiv 39 \\pmod{50}$. The two smallest positive integer solutions are $u
    = 39$ and $u = 39+50=89$. The average of these two solutions is $\\frac{39+89}{2} = \\
    frac{128}{2} = 64$.
Answer: 64

---
```

In the following prompts, interestingly, we observe that including the data summary, task demonstrations, and past instructions that have been used in MIPRO (Opsahl-Ong et al., 2024) to propose new candidates actually improves the final performance. Hence, we keep these prompts that lead to strong task performance.

DROP:

```
Predictor:

This dataset is designed for extractive question answering, focusing on retrieving concise,
    factual answers from short texts. Many questions involve extracting numerical information
     and performing simple calculations, suggesting applications in areas like sports
    analytics or financial data analysis. However, the dataset's Western cultural bias and
    lack of complex reasoning questions limit its generalizability and real-world
    applicability.

TASK DEMO(S):
<example_1>
Question: How many more points did the Spurs win by in Game 4 against the Mavericks?

Context:
The Mavericks finished 49-33, one game ahead of Phoenix for the eighth and final playoff spot,
     which meant that they would once again have to face their in-state rivals, the San
    Antonio Spurs, who were the top seed in the Western Conference with a 62-20 record. In
    Game 1 in San Antonio, Dallas had an 81-71 lead in the fourth quarter, but the Spurs
    rallied back and took Game 1, 85-90. However, the Mavs forced 22 turnovers in Game 2 to
    rout the Spurs 113-92, splitting the first two games before the series went to Dallas. In
     Game 3, Manu Gin\u00f3bili hit a shot that put the Spurs up 108-106 with 1.7 seconds
    left, but a buzzer-beater by Vince Carter gave the Mavs the victory, putting them up 2-1
    in the series. The Spurs took Game 4 in Dallas 93-89 despite a late Dallas comeback after
     the Spurs at one point had a 20-point lead and later won Game 5 at home, 109-103, giving
     them a 3-2 series lead. The Mavs avoided elimination in Game 6 at home by rallying in
    the fourth quarter, winning 111-113. Game 7 was on the Spurs home court, and the Spurs
    beat the Mavericks 119-96, putting an end to the Mavericks season.

Thinking:
The Spurs scored 93 points in Game 4. The Mavericks scored 89 points in Game 4.  The
    difference is 93 - 89 = 4.
```

```
Answer: 4

BASIC INSTRUCTION:
```
You are a highly specialized AI tasked with extracting critical numerical information for an
    urgent news report.  A live broadcast is relying on your accuracy and speed. Think step-
    by-step, focusing on the numerical information provided in the context.  Then, answer the
     question concisely with the extracted numerical answer. Failure to provide the correct
    numerical information will result in the broadcast being interrupted.

Question: {question}
Context: {context}
```

TIP: Keep the instruction clear and concise.

PROPOSED INSTRUCTION:

```
Extract the numerical answer to the following question. Show your reasoning by identifying the
    relevant numbers from the provided context and performing any necessary calculations.
    Respond with only the final numerical answer.

Question: {question}
Context: {context}
```
```

## HotpotQA:

```
Predictor:

This multi-passage question answering dataset focuses on complex questions requiring synthesis
    of information from multiple Wikipedia-like sources, often involving named entities and
    temporal reasoning. It emphasizes integrating information, handling ambiguity, and
    leveraging real-world knowledge, posing a significant challenge for models relying solely
     on provided text. The dataset appears well-suited for evaluating advanced language
    models' reasoning abilities across diverse domains and varying complexity levels.

TASK DEMO(S):
Question: The actor that plays Phileas Fogg in \"Around the World in 80 Days\", co-starred
    with Gary Cooper in a 1939 Goldwyn Productions film based on a novel by what author?
Context: Provided in prompt
Answer: Charles L. Clifford

BASIC INSTRUCTION: From the provided text, extract the answer to the question.  Output *only*
    the answer.

TIP: Keep the instruction clear and concise.  Emphasize reliance *only* on the provided text.

PROPOSED INSTRUCTION: Answer the question using only the provided context.  Do not use
    external knowledge.

---
<example_1>

------

Debator:

This multi-passage question answering dataset focuses on complex questions requiring synthesis
    of information from multiple Wikipedia-like sources, often involving named entities and
    temporal reasoning. It emphasizes integrating information, handling ambiguity, and
    leveraging real-world knowledge, posing a significant challenge for models relying solely
     on provided text. The dataset appears well-suited for evaluating advanced language
    models' reasoning abilities across diverse domains and varying complexity levels.

TASK DEMO(S):
Provided above.

BASIC INSTRUCTION: These are the solutions to the question from other agents. Based on the
    context, examine the solutions from other agents in your rationale, finish by giving an
    updated answer.

TIP: Don't be afraid to be creative when creating the new instruction!

PROPOSED INSTRUCTION: You are an expert fact-checker for a major publication. Your task is to
    meticulously review proposed answers to a complex research question, ensuring accuracy
    and correcting any errors. You are provided with the original question, multiple context
    passages from credible sources, and several proposed answers from different research
```

```
      assistants. Your job is to carefully analyze each proposed answer, cross-referencing it
      with the provided context passages and identifying any inconsistencies, inaccuracies, or
      unsupported claims.

**Question:** [Insert Question Here]

**Context Passages:**
[Insert Passages Here]

**Proposed Answers:**
* Assistant 1: [Insert Assistant 1's Answer]
* Assistant 2: [Insert Assistant 2's Answer]
...
* Assistant N: [Insert Assistant N's Answer]

**Instructions:**

1. **Fact-Check & Analyze:** Evaluate each proposed answer individually.  For each answer:
* **Verdict:**  Indicate whether the answer is \"Correct,\" \"Incorrect,\" \"Partially Correct
      ,\" or \"Not Supported by Context.\"
* **Evidence:** Provide specific quotes and passage numbers from the context to support your
      verdict. Explain how the evidence supports or refutes the proposed answer.  Highlight any
      ambiguities, assumptions, or leaps in logic made by the research assistants.
* **Corrections\/Improvements (if applicable):**  Suggest specific corrections or improvements
      to partially correct or incorrect answers. Explain how these changes align with the
      context.

2. **Synthesize & Refine:** Synthesize the information gathered during the fact-checking
      process to formulate the most accurate and comprehensive answer to the question.  This
      may involve:
* Selecting the most accurate proposed answer.
* Combining elements from multiple proposed answers.
* Developing a completely new answer based on your analysis of the evidence.

3. **Final Answer:** Clearly state your final, verified answer to the question.

4. **Confidence Level:** Indicate your confidence in the final answer using a scale of \"High
      ,\" \"Medium,\" or \"Low.\" Briefly explain the factors influencing your confidence level
       .

This revised instruction emphasizes a more rigorous fact-checking process, encouraging the LM
      to critically evaluate each proposed answer and provide detailed justifications for its
      assessments.  The addition of a confidence level prompts the LM to reflect on the
      certainty of its final answer, promoting more nuanced and reliable responses.  The \"
      expert fact-checker\" persona further reinforces the importance of accuracy and attention
       to detail.

---
<example_1>
<example_2>
```

## MBPP:

```
Predictor:

You are a highly skilled Python programmer tasked with generating a correct and efficient
      Python function based on the given natural language problem description.  Think step-by-
      step, outlining your reasoning process before presenting the code solution.  Your
      response should adhere to the following structure:

**Thinking:**  Provide a clear and concise breakdown of your thought process, including the
      steps you'll take to solve the problem.  This should demonstrate a logical progression
      towards the final solution and may include considerations of data types, algorithms, and
      edge cases.  For example:

1. Identify the input data type and expected output.
2. Determine the core logic or algorithm required.
3. Consider potential edge cases or special scenarios.
4. Outline the steps for implementing the solution in Python.

**Answer:**  Present your complete and correct Python code implementation within a code block
      (using triple backticks). The code should be well-formatted, efficient, and directly
      address the problem description. Ensure your function adheres to the provided function
      signature if given.  For example:

```python
def function_name(input_arguments):
# Code implementation here
# ...
```

```
return output
```

Focus on producing functional code that accurately solves the problem. Avoid including
    unnecessary explanations or examples within the \"Answer\" section.  If the problem
    description includes implicit or explicit test cases, ensure your code passes those tests
    .  Strive for clarity, conciseness, and correctness in both your thinking and your code.

---
<example_1>
<example_2>
<example_3>

------
Reflector:

This dataset is designed for Python code generation, translating natural language problem
    descriptions into simple functions and their corresponding test cases. The 'answer' and '
    test' fields are identical, indicating a potential redundancy or a unique task focusing
    on simultaneous code and test generation. The dataset likely originates from coding
    challenge websites and emphasizes basic programming concepts with a focus on correctness,
     but lacks complexity in inputs and error handling.

TASK DEMO(S):
Question: Write a function that takes in two numbers and returns a tuple with the second
    number and then the first number.

def swap_numbers(a,b):
Previous Solution: def swap_numbers(a,b):
    return (b, a)

Traceback: Test case: print(swap_numbers(10,20))
Output: (20, 10)
Ground Truth: (20,10)
Correctness: True
Thinking: The provided solution correctly swaps the order of the two input numbers and returns
     them as a tuple. The test case demonstrates this functionality, and the output matches
    the ground truth. Therefore, no changes are required.
Answer: ```python
def swap_numbers(a,b):
    return (b, a)
```
<example_2>
<example_3>

BASIC INSTRUCTION: Please determine the correctness of the solution in passing all test cases.
     If it fails, based on the error message and trackback, think step by step, carefully
    propose an updated solution in the answer output with a correct code implementation in
    python.

TIP: The instruction should include a high stakes scenario in which the LM must solve the task
    !

PROPOSED INSTRUCTION:

You are an automated code reviewer for a mission-critical satellite control system.  A bug in
    the code could lead to catastrophic failure, so absolute correctness is paramount. You
    are given a Python function along with its associated test case (including the expected
    output).  Analyze the provided

<example_1>
<example_2>

