# OpenReview forum: "Multi-Agent Design: Optimizing Agents with Better Prompts and Topologies"
_ICLR.cc/2026/Conference — ICLR 2026 Poster_

### Official Review · Reviewer_NqFK · 2025-10-29

**Soundness:** 2
**Presentation:** 3
**Contribution:** 2
**Rating:** 4
**Confidence:** 4

**Summary:**

This paper presents Multi-Agent System Search (MASS), a novel three-stage framework to automate the complex design of multi-agent systems by optimizing both prompts and interaction topologies. The contribution is an interleaved optimization process that first locally optimizes individual agent prompts, then uses those insights to search a pruned topology space, and finally performs a global prompt optimization on the best-found workflow. The experiments are extensive and show that this method substantially outperforms current SOTA baselines.

**Strengths:**

1.	Novel and Effective Framework: The proposed three-stage MASS framework is a novel and intuitive contribution. It effectively decomposes the highly complex joint-optimization problem of prompts and topologies into three manageable stages.

2.	Strong Empirical Performance: The paper demonstrates substantial and consistent performance gains across a diverse set of 8 benchmarks, significantly outperforming existing manual MAS designs and state-of-the-art automated frameworks.

3.	Good Presentation: The paper is well-written, clearly structured, and easy to follow, which effectively communicates the authors' core contributions and experimental results.

**Weaknesses:**

1.	The core three-phase framework of this paper is inspired by two preliminary analyses, which seem to have methodological flaws. The "prompt-first" conclusion shown in Figure 2 has not been strictly proven, as it lacks a key benchmark to compare it with the "topology-first" method. Moreover, it is unclear whether the results shown in Figure 4 are based on the isolated evaluation of components or the ablation of completed topology. If it is the former, it ignores the synergistic nature of the MAS topology, where components only show value when combined.

2.	The proposed framework is presented as a heuristic and lacks theoretical grounding. The three-stage decomposition (local prompt, topology search, global prompt) is justified by empirical observation rather than a formal analysis of the optimization space.

3.	The paper's presentation suffers from minor issues that harm readability. There are spelling errors ("Monte-Carto" in the appendix) and the "Related Work" section is placed between the Method and Experiment sections, which disrupts the logical flow of the paper.

**Questions:**

1.	The paper convincingly shows that “Prompting to SC” is effective, but it does not include the critical baseline of “SC to Prompting”. How to determine the optimal order of optimization directives before topology optimization, rather than the opposite?

2.	Could the authors provide a more detailed analysis of the framework's scalability? Specifically, how does the optimization cost of Stage 1 (1PO) scale with the number of available topology building blocks, and how does the cost of Stage 3 (3PO) scale with the complexity (e.g., number of nodes) of the final discovered topology?

---

> ### Author Response · Authors · 2025-11-21
> **Response to Reviewer NqFK (Part 1/2)**
>
> We thank the reviewer for their constructive feedback and acknowledging the novelty and strong empirical performance of MASS! We include the insightful topology-first experiment and a detailed analysis of the framework scalability in the response. We hope that our response sufficiently addresses the reviewer’s remaining concerns, and the reviewer could consider improving their score.
>
> > The core three-phase framework of this paper is inspired by two preliminary analyses, which seem to have methodological flaws. The "prompt-first" conclusion shown in Figure 2 has not been strictly proven, as it lacks a key benchmark to compare it with the "topology-first" method.
>
> We would like to clarify that we did not claim that PO has to happen before TO. Instead, MASS is an optimization framework that resolves the complexity of joint optimization via interleaving the optimization stages between the PO and TO, and MASS also includes the workflow-level prompt optimization (3PO) **after** 2TO to specifically model the global dependence across agents. Hence, the topology-first approach is a sub-component of the MASS framework, and 1PO only serves as a warm-up phase, guaranteeing each agent is primed for their roles, and ensuring a fair evaluation in the topology optimization stage.
>
> In the context of Section 2.1, we deliver the finding - *optimize agents locally before scaling their topology*, which is under a strict constraint, where we only consider agents within the same type of building block (e.g., SC or Reflect) without the synergy with other modules. This is driven by the conditional dependence of workflow $\mathcal{W}(a_i|p_i)$, where $p_i$ represents the individual prompt per agent $a_i$. Conditioning on ill-defined prompts will result in undesirable functionality and ineffective scaling behavior with potential error amplification in MAS, which is further reflected in Table 4 (the Claude-3.5-Sonnet experiment), where the debate topology with manual prompts is initially ill-conditioned, and scaling that topology only has a decremental system-level effect.  We agree with the reviewer that “topology-first” is an insightful experiment, and we include it in the following response.
>
> > The paper convincingly shows that “Prompting to SC” is effective, but it does not include the critical baseline of “SC to Prompting”. How to determine the optimal order of optimization directives before topology optimization, rather than the opposite?
>
> The topology-first approach enables the follow-up PO to model the global dependence across agents. Given “SC to Prompting” as an example, the SC or the Aggregate block typically benefits from the diversity of predictions across agents, and conducting PO on top of SC will also improve the diversity of predictions, thereby improving the final performance from 83.0% to 84.6% over the “Prompting to SC” approach. However, it is worth noting that the main computation overhead comes from the prompt optimizer that iteratively evaluates a pool of instruction candidates. Consequently, the topology-first approach will result in $\mathcal{N}(a_{\text{SC}})$ times of inference cost. Therefore, driven by the distribution of search cost, MASS is constructed via both the prompt-first (1PO->2TO) and topology-first approach (2TO->3PO) in a most manageable search budget via a local-to-global optimization manner.
>
> | Method | MATH (%) | API Calls for PO (Val) |
> |----------|-------|-------|
> | Prompting->SC  | 83.0 | 100 |
> | SC->Prompting  | 84.6 | 900 |
>
> > It is unclear whether the results shown in Figure 4 are based on the isolated evaluation of components or the ablation of completed topology. If it is the former, it ignores the synergistic nature of the MAS topology, where components only show value when combined.
>
> Figure 4 is evaluated based on the *independent* topology component, providing an insightful finding that some building blocks may not be beneficial to the MAS performance. Inspired by that, we introduce an incremental influence function $I$ that quantifies the relative importance ratio of the current search dimension. It downweights the selection probability of low-performing building blocks, thereby improving the cost-effectiveness of the overall search process. *The synergy of low-performing modules and their combination will still be evaluated* in 2TO, and Figure 5 (Right) further validates the effectiveness of the pruning process. The smoothness of the softmax function guarantees the theoretical global optimality in the topology optimization, and the interdependence between modules will be further optimized in the third phase of workflow-level optimization.

---

> ### Author Response · Authors · 2025-11-21
> **Response to Reviewer NqFK (Part 2/2)**
>
> > The proposed framework is presented as a heuristic and lacks theoretical grounding. The three-stage decomposition (local prompt, topology search, global prompt) is justified by empirical observation rather than a formal analysis of the optimization space.
>
> We agree with the reviewer that the framework is motivated by empirical observations, the complexity of the search space, and the conditional dependence between agent topology and their prompts. However, along with the development of sophisticated prompt optimization methods, this research field is primarily driven by empirical evidence and brings many practical, influential impacts. Though it will be good to have a theoretical grounding, our approach should not be penalized for a shortcoming of the entire research field.
>
> > The paper's presentation suffers from minor issues that harm readability. There are spelling errors ("Monte-Carto" in the appendix) and the "Related Work" section is placed between the Method and Experiment sections, which disrupts the logical flow of the paper.
>
> We thank the reviewer for suggesting a better flow of narratives. The previous related work section presents a common placement, providing readers with analysis and the main contents more directly. In light of the suggestions from the reviewer, we have updated the manuscript, fixed any typos, and replaced the related work section (highlighted in Magenta) for a better logical flow of the paper in the updated manuscript.
>
> > Could the authors provide a more detailed analysis of the framework's scalability? Specifically, how does the optimization cost of Stage 1 (1PO) scale with the number of available topology building blocks, and how does the cost of Stage 3 (3PO) scale with the complexity (e.g., number of nodes) of the final discovered topology?
>
> We thank the reviewer for suggesting a scalability analysis with respect to the growing number of building blocks and the number of agents in the system. To estimate the computation cost for each stage of MASS, we provide the following formulation: 1) First, the prompt optimization will evaluate the MAS with $K$ rounds of evaluation, where each round consists of $M$ prompt candidates, and each system inference requires $\mathcal{N}(a)$ API calls by the number of agents in the MAS. Hence, 1PO scales linearly with the number of building blocks $J$, and its cost can be computed as $C(\text{1PO}) = \sum_j^{J} \mathcal{N}(a_j) \times M \times K$; 2) The topology optimization evaluates $N$ topology candidates, such that $C(\text{2TO}) = \sum_n^{N} \mathcal{N}(\mathcal{W}_n)$; 3) Similar to (1), 3PO conducts PO on the best-found workflow, such that $C(\text{3PO}) = \mathcal{N}(\mathcal{W}^*)\times M \times K$. Therefore, the cost of 3PO also scales linearly with the number of nodes in the best-found topology. We have updated the cost analysis section (Appendix D.3) in the manuscript with this valuable new analysis.

---

> > ### Author Response · Authors · 2025-12-03
> > **Summary of response**
> >
> > We'd like to thank the reviewer again for their constructive review and feedback. To our understanding, the reviewer was primarily concerned about (1) a topology-first experiment and (2) a scalability analysis of MASS -- we believe that we have addressed each of the concerns thoroughly:
> > 1. We clarified that the topology-first approach is already a sub-component of MASS. In light of the reviewer's suggestion, we included a new experiment comparing both prompt-first and topology-first approaches with their costs, which further demonstrates the cost-effectiveness of MASS in a local-to-global order of optimization.
> >
> > 2. We provided a detailed cost analysis of the scalability of MASS in equations, where we show that the cost of MASS only scales linearly with the number of agents.
> >
> > We hope that our response sufficiently addresses the reviewer’s remaining concerns.

---

### Official Review · Reviewer_ib6h · 2025-10-30

**Soundness:** 3
**Presentation:** 3
**Contribution:** 2
**Rating:** 4
**Confidence:** 4

**Summary:**

This paper proposes Multi-Agent System Search (MASS), an optimization framework that seeks the optimal multi-agent system. It models a workflow as a collection of building blocks and their logical relationships, and constructs a better MAS through a three-stage process: first optimizing prompts at the building block level, then optimizing the workflow topology, and finally refining the global prompt.

**Strengths:**

- This paper jointly considers prompt and topology optimization, and alternately updates both components, offering a novel optimization strategy for MAS.
- The building block design decouples prompt optimization from topology optimization, effectively reducing the cost of searching for an effective MAS structure by pruning the search space.
- Experimental results demonstrate performance improvements compared to baselines.

**Weaknesses:**

- Inconsistent notations. In Equation (1), the symbol $a$ is defined as a configuration. However, later in the text, $a$ is ambiguously reused to refer to individual agents and topologies.
- Figure 4 evaluates individual topologies in isolation on specific tasks. However, the final MAS in this work is a composition of multiple topologies. It is unclear whether these isolated results sufficiently support the claim that ``not all topologies have a positive influence on MAS design.'' Similarly, during search space pruning, the selection criterion is based on the performance of each building block relative to an initial agent, rather than the performance of the full composite structure. This raises a concern: Could this approach inadvertently prune away building blocks that are suboptimal in isolation but essential for achieving global optimality in combination?
- The proposed pipeline first performs prompt optimization on all building blocks before search space pruning. Given that some of these blocks may ultimately be discarded during pruning, this may incur potentially unnecessary computational overhead.

**Questions:**

- How does the evaluator assign scores? How is the validation set determined, and how is the validation performance computed?
- The description in lines 260–269 is confusing, largely due to the ambiguous reuse of the symbol $a$. Could you provide a clear and precise explanation of this section?

---

> ### Author Response · Authors · 2025-11-21
> **Response to Reviewer ib6h (Part 1/2)**
>
> We thank the reviewer for their constructive feedback, which helps to strengthen our manuscript! We provide clarification on the notation, a detailed discussion on the theoretical optimality of MASS optimization, and the necessity of 1PO in our framework, where we have also updated the manuscript accordingly. We hope that our response sufficiently addresses the reviewer’s remaining concerns, and the reviewer could consider improving their score.
>
> > Inconsistent notations. In Equation (1), the symbol $a$ is defined as a configuration. However, later in the text,  $a$ is ambiguously reused to refer to individual agents and topologies.
>
> We would like to clarify that $a$ is consistently represented as the agentic configuration for a MAS system in our work, where individual agents and the topology can both be parameterized via specific configurations. Given a workflow $\mathcal{W}(a)$ over the building blocks, the overall search space forms a supernet of agents. For instance, given a simple search space $\mathcal{A}=\[a_0, a_1\]$ of [Aggregate, Reflect], a single agent system ($a_0$) in this search space can be parameterized as [1, 0], indicating that there is only a single reasoning agent, while other building blocks are deactivated. This representation then generalizes to more complex topologies, and we kindly refer the reviewer to Figure 8 and Table 2 for more visualization of different configurations. Following the suggestion from the reviewer, we also updated the manuscript, Section 2, with a clarification on the connection between the configuration to specific agents and topologies, and updated the terms in Section 3.2 accordingly.
>
> > Could this approach inadvertently prune away building blocks that are suboptimal in isolation but essential for achieving global optimality in combination?
>
> The main idea of the topology optimization with pruning (2TO) is to downweight the selection probability of low-performing building blocks when we form the search space per candidate (Line 10-11, Algorithm 1), thereby improving the cost-effectiveness of the overall search process. This selection probability $p$ is controlled by a softmax function that is inherently *smooth*. Hence, the combination of low-performing modules will still be evaluated, but at a lower probability weighted by the selection probability. It guarantees the theoretical global optimality given a sufficiently large number of samples, and the interdependence between modules will be further optimized in the third phase of workflow-level optimization. Practically, we notice that it is also empirically unlikely for low-performing modules to suddenly become better when combined together (see Figure 5 (Right) box plot of ‘w/o Prune’ with a lower maximum point), which is also a common practice in neural architecture search (NAS) to prune low-influence, redundant features that do not harm the global composite performance [1].
>
> [1] Wan, X., Ru, B., Esperança, P. M., & Li, Z. (2022). On Redundancy and Diversity in Cell-based Neural Architecture Search. ICLR 2022.
>
> > The proposed pipeline first performs prompt optimization on all building blocks before search space pruning. Given that some of these blocks may ultimately be discarded during pruning, this may incur potentially unnecessary computational overhead.
>
> In MAS, the performance of individual building blocks is first conditioned on their prompts, and it is practically challenging to initiate well-functioning building blocks via manual engineering alone. Consequently, any ill-defined block without prompt optimization will result in an inaccurate evaluation of its incremental influence ratio and compositional performance in the phase of 2TO. Therefore, carrying out 1PO to every block guarantees a fair evaluation in the topology optimization and is necessary in the overall framework. In addition, we conduct 1PO on each building block with a minimum number of agents (listed in Table 3) to maintain their functionality, which effectively reduces the computational overhead in this early process.

---

> ### Author Response · Authors · 2025-11-21
> **Response to Reviewer ib6h (Part 2/2)**
>
> > How does the evaluator assign scores? How is the validation set determined, and how is the validation performance computed?
>
> We kindly refer the reviewer to Appendix C.1 for the detailed specification of the validation set, where we randomly sample a fixed number of labelled samples for validation. The evaluation for MAS is therefore computing metrics (e.g., Accuracy, F1, and pass@1) for the system performance per dataset.
>
> > The description in lines 260–269 is confusing, largely due to the ambiguous reuse of the symbol $a$. Could you provide a clear and precise explanation of this section?
>
> Following a consistent definition of $a$ as the agentic configuration and $a_i$ representing each search dimension as edited in the updated manuscript, Section 2, descriptions in Lines 260-269 correspond exactly to Algorithm 1, Lines 10-16. The incremental influence function $I$ first quantifies the relative importance ratio of the current search dimension $a_i$ compared to the base operator, which then assigns a probability for controlling the activation of each search dimension $a_i$ (Line 10). For each sampling of a new candidate (Line 11), the search space is constructed based on the activation status of each search dimension $\mathcal{A}=$ {$a_i$} if $u_i$ < $p_{a_i}$. The final configuration candidate is then randomly sampled from the current search space $a \sim \mathcal{A}$, which leads to the complete workflow for evaluating the MAS candidate by $\mathcal{W}(a)$. We hope this response addresses any remaining concerns from the reviewer.

---

> > ### Comment · Reviewer_ib6h · 2025-11-26
> >
> > Thank you for your responses. Most of my concerns have been addressed, and I would like to raise my score. I suggest that the presentation of this paper should be carefully polished for better understanding.

---

> ### Author Response · Authors · 2025-11-29
>
> Thank you for your feedback and for raising the score! We are glad that our response has addressed the reviewer's concerns. We have incorporated the feedback from the reviewer and updated the manuscript with the improvements in presentation in the latest revision.

---

### Official Review · Reviewer_P2XL · 2025-11-01

**Soundness:** 4
**Presentation:** 4
**Contribution:** 3
**Rating:** 6
**Confidence:** 3

**Summary:**

This paper formalizes the Multi-Agent-System into a complex problem of joint optimization prompts and topology, and proposes the MASS framework to efficiently search in the MAS design space. The framework is divided into three parts, block-level prompt optimization, workflow topology optimization, and workflow-level prompt optimization, and the author claims that this method outperforms existing MAS in multiple tasks.

**Strengths:**

The paper presents a clear and logical narrative process. It explicitly formalizes the design of MAS as a joint optimization problem of prompts and topology, introduces an incremental impact metric to prune the huge space, and demonstrates the significance of this consideration through detailed experiments.
The experiments show the optimization trajectory while taking into account cost-effectiveness. MASS is a plug-and-play module that provides a platform for future expansion and promotes MAS from manual trial and error to automated search.

**Weaknesses:**

The MASS framework mentions that the topology search in the second stage is guided by the "incremental influence" indicator, which evaluates each module independently and cannot find synergies between dependent modules.
This "exhaustive task" can be performed by some more advanced algorithms, such as evolutionary algorithms, and the operation of this random search can be more fully demonstrated according to the change of performance scores, and the potential optimization direction in the future can also be more complete.
This framework is theoretically highly sensitive to temperature parameters, and it is helpful to analyze on temperature settings.

**Questions:**

How does the MASS consider the combination order of topology modules, and is the current MASS framework effective for the combination order of different topology modules? Since the authors already want to use the random search strategy to select the number of modules, can we put the combination order of modules into the search space to realize the automatic combination of different tasks?

---

> ### Author Response · Authors · 2025-11-21
> **Response to Reviewer P2XL**
>
> We thank the reviewer for their insightful comments and valuable suggestions towards future developments! We hope that our response sufficiently addresses the reviewer’s remaining concerns, and the reviewer could consider improving their score.
>
> > The topology search in the second stage is guided by the "incremental influence" indicator, which evaluates each module independently and cannot find synergies between dependent modules.
>
> We agree with the reviewer that our influence function treats modules independently, where the sampling probability of low-performing modules will be downweighted. This indicator leads to a more effective search space for MAS optimization. Meanwhile, the *synergy* between different modules will still be evaluated in our topology optimization, but at a lower frequency defined by the influence indicator, and their interdependence will be optimized later in the third phase of workflow-level optimization. In addition, we would like to highlight that the MASS optimization is carried out in a novel context with agentic building blocks, where each building block represents a unique functionality in MAS, including actions like summarization and reflection. The performance of these distinct building blocks is less correlated with each other.
>
> > This "exhaustive task" can be performed by some more advanced algorithms, such as evolutionary algorithms, and the operation of this random search can be more fully demonstrated according to the change of performance scores, and the potential optimization direction in the future can also be more complete.
>
> We appreciate the suggestions from the reviewer for algorithms such as the evolutionary algorithm. Given the complexity of the existing search space, our topology optimizer and the dynamic search space pruning process in MASS have provided solid empirical evidence for efficiently discovering optimal topologies in a manageable cost budget. In terms of future MAS design with a growing complexity of the problem, we have also considered more advanced algorithms like the Bayesian optimization [1] in the section of Future work (Lines 936 - 938), which can come with a better sample efficiency while being capable of modelling the interdependence between the features. Since the random search with our effective search space pruning is already a strong approach, extending MASS to more advanced algorithms like Bayesian optimization will only perform strictly better, which is also complementary to our proposed search space pruning process. Therefore, instead of a limitation of our approach, it actually reveals the advantage of MASS with plug-and-play extension space for the topology optimizer to conquer more complex future MAS designs.
>
> [1] Kandasamy, K., Neiswanger, W., Schneider, J., Poczos, B., & Xing, E. P. (2018). Neural architecture search with bayesian optimisation and optimal transport. NeurIPS 2018.
>
> > This framework is theoretically highly sensitive to temperature parameters, and it is helpful to analyze temperature settings.
>
> In Figure 5 (Right), we provided an ablation study on the topology optimization with and without pruning. Since the chance of pruning is controlled by the selection probability via a softmax function of the influence $I$, TO without pruning is therefore equivalent to a relatively large temperature (i.e., $t>1$), which leads to a roughly uniform distribution given that the incremental influence $I$  is typically close to 1. TO with pruning has its temperature $t=0.05$, which allows us to sharpen the probability distribution and actively prunes out dimensions. From the box plot, it is observable that high temperature will result in a larger interquartile range with a lower maximum point. Our low-temperature control leads to a more cost-effective search in the pruned search space.
>
> > How does the MASS consider the combination order of topology modules, and is the current MASS framework effective for the combination order of different topology modules? Can we put the combination order of modules into the search space to realize the automatic combination of different tasks?
>
> We include the detailed construction rule in Appendix C.3. Throughout our study, we find that the impact of different agentic configurations outweighs the ordering substantially. Therefore, our rule follows the practice of agent designs in a predefined fixed order. We agree with the reviewer that MASS is fully feasible and has more potential to treat the topology order as an additional search dimension, thereby fully automating the complete MAS design. In that case, the agentic configuration will be conditioned on the order, and the search dimension will grow exponentially with the number of building blocks. This setup will be in favor of more advanced search algorithms with conditional dependence modelling, where Bayesian optimization can be a desirable algorithm candidate.

---

### Official Review · Reviewer_TPid · 2025-11-03

**Soundness:** 3
**Presentation:** 3
**Contribution:** 3
**Rating:** 6
**Confidence:** 4

**Summary:**

This paper proposes MASS (Multi-Agent System Search), a framework for automatically optimizing multi-agent systems (MAS) by jointly optimizing both prompts and topologies. The authors first conduct an analysis showing that prompt optimization and topology design are critical factors for MAS performance. Based on these insights, MASS employs a three-stage optimization approach: (1) block-level prompt optimization for individual agent types, (2) workflow topology optimization in a pruned search space, and (3) workflow-level prompt optimization for the complete system.

**Strengths:**

1, Comprehensive Analysis: The paper provides valuable insights into what makes effective MAS through systematic ablations (Sec 2.1-2.2), showing that prompts matter more than previously recognized and that only a small fraction of topologies are beneficial.

2, Well-Motivated Design: The three-stage optimization approach (local→global, prompts→topologies) is intuitive and addresses the complexity of joint optimization effectively. The influence-weighted search space pruning (Eq. at line 269) is a practical contribution.

3, Strong Empirical Results: MASS achieves substantial improvements across diverse tasks (78.79% avg on Gemini Pro vs 70.26% for best baseline). The results are consistent across multiple LLM backbones (Gemini, Claude, Mistral).

**Weaknesses:**

1, Limited Novelty: The core contribution is primarily engineering—combining existing prompt optimization (MIPRO) with topology search. The individual components (APO, topology search) are not novel. The main novelty lies in the specific combination and the three-stage approach.

2, Search Space Design Limitations:
- The topology space is limited to 5 predefined building blocks. More complex or novel topologies are not explored.
- The "rule-based" construction order (line 275-276) seems arbitrary and is not well justified. Why is [summarize, reflect, debate, aggregate] the right order?
- The search space is task-specific (Table 2), limiting generalizability.

3,Methodological Concerns:
- AFlow comparison: The authors acknowledge AFlow uses Claude 3.5 as optimizer while Gemini as executor, making comparison unfair (marked with *). This is a significant limitation.
- Statistical significance: While error bars are shown, no formal significance tests are provided.
- Validation/test split: Using only 50-100 examples for validation seems small and may lead to overfitting to the validation set.

4, Incomplete Analysis:
- Why does prompt optimization help so much? The paper shows that it does (Fig 2) but provides limited insight into what makes optimized prompts better.
- Topology analysis: Fig 4 shows different topologies work for different tasks, but there's limited discussion of why. What task characteristics favor which topologies?
- Workflow-level optimization gains: Stage 3 provides only ~2% improvement (line 342). Is this consistent? When does it help?

**Questions:**

See weakness

---

> ### Author Response · Authors · 2025-11-21
> **Response to Reviewer TPid (Part 1/2)**
>
> We thank the reviewer for their insightful comments and acknowledging the merits of our analysis and empirical improvements! We provide further clarifications regarding the reviewer’s concerns on both the methodology and the analysis. We hope that our response sufficiently addresses the reviewer’s remaining concerns, and the reviewer could consider improving their score.
>
> > 1. Limited Novelty: The core contribution is primarily engineering—combining existing prompt optimization (MIPRO) with topology search. The individual components (APO, topology search) are not novel. The main novelty lies in the specific combination and the three-stage approach.
>
> We would like to argue that the novelty of MASS is beyond simply combining a prompt optimizer with a topology optimizer. The core contribution of MASS lies in overcoming the complexity of MAS optimization via interleaved optimization stages, where each stage comes with a **tractable** search space, which is extremely important considering the cost of optimizing real-world MAS in terms of a joint optimization space for both prompts and topologies. Our method involves a novel block-level prompt optimization and a dynamic pruning process in topology optimization for effective search space pruning. The complete three-stage optimization not only conquers the optimization complexity but also models cross-agent dependence, which is not previously common, via a workflow-level prompt optimization. The overall framework is plug-and-play with any off-the-shelf prompt optimizer and generalizable to a customizable search space. In addition, we deliver novel findings on the impact of prompt optimization on MAS and the redundancy in the traditional MAS search space, which have been largely overlooked by the existing community and will bring substantial benefits to future agent research towards building more efficient agentic systems.
>
> > 2.1. The topology space is limited to 5 predefined building blocks. More complex or novel topologies are not explored;
> 2.3. The search space is task-specific (Table 2), limiting generalizability.
>
> We would like to point out that the proposed MASS framework, as listed in Algorithm 1,  is a general MAS optimization framework applicable to any search space $\mathcal{A}$ with configurable search dimensions $a_i$. Our experiments have investigated serial (Reflect), parallel (Aggregate), and a mixture of connection (Debate) topologies, which are practically common, effective building blocks that are shared across all tasks in Table 2. The search space can also be easily extended with any custom blocks, where we have shown the general feasibility of MASS with “tool-use” and “summarize” as additional search dimensions. It is worth noting that the MAS optimization always comes with a trade-off between the dimension of the search space and the effectiveness of its subspace. For instance, ADAS searches within an unbounded, LLM-proposed topology space. However, ADAS easily gets trapped in over-complex, low-performant topologies (see Figure 6), whereas MASS optimizes novel prompts in a more effective search space. Overall, as discussed in the limitation section (Lines 930-935), we agree with the reviewer that incorporating other topologies may further improve the final performance of MASS, which is complementary to the development of MASS, and we will endeavor to explore this in future work.
>
> > 2.2. The "rule-based" construction order (line 275-276) seems arbitrary and is not well justified. Why is [summarize, reflect, debate, aggregate] the right order?
>
> We kindly refer the reviewer to Appendix C.3, where we discussed the detailed construction rules. Throughout our study, we find that the impact of different agentic configurations (e.g., the number of agents in parallel) outweighs the ordering substantially, where exchanging the position of “reflect” and “debate” only has a marginal performance difference. Though our MASS framework is fully capable of incorporating ordering as a distinct search space, it will inevitably introduce redundancy into the final search space, resulting in less-efficient search performance, and the search cost grows exponentially. Hence, our rule follows a simple practice of agent design patterns by constructing topology, such that the relevant context is first summarized, refined via self-reflection and debating, and finally aggregated via majority voting.

---

> > ### Author Response · Authors · 2025-11-21
> > **Response to Reviewer TPid (Part 2/2)**
> >
> > > 3.1. AFlow comparison: The authors acknowledge AFlow uses Claude 3.5 as optimizer while Gemini as executor, making comparison unfair (marked with *). This is a significant limitation.
> >
> > We would like to recall that AFlow relies on an LLM-based optimizer for node expansion, where the meta-prompt of its optimizer is specifically built upon Claude-3.5-Sonnet and cannot generalize to other LLM families with its designed functionality. It is also worth noting that the general reasoning performance of Claude-3.5-Sonnet is actually on par and even outperforms Gemini 1.5 Pro in benchmarks. Therefore, the MASS experiments are not taking any advantage of using a stronger LLM-based optimizer over the baseline, and we ensure a fair comparison with the Gemini-1.5 as the common executor.
> >
> > > 3.2. Statistical significance: While error bars are shown, no formal significance tests are provided.
> >
> > We thank the reviewer for suggesting the significance test. In Table 1, we compute that all empirical results by MASS with performance improvements over the baselines have consistent p-value < 0.05, which indicates the statistical significance of our results.
> >
> > > 3.3. Validation/test split: Using only 50-100 examples for validation seems small and may lead to overfitting to the validation set.
> >
> > The relatively small validation set actually reflects the sample efficiency of the MASS framework, which is one of the unique advantages of our method. It shows that MASS can effectively optimize both prompts and topologies in a complex search space by leveraging a limited amount of data, and additional data samples can only improve the performance of MASS.
> >
> > > 4.1. Why does prompt optimization help so much? The paper shows that it does (Fig 2), but provides limited insight into what makes optimized prompts better.
> >
> > LLM-based agents have been shown to exhibit severe sensitivity to their prompt designs. Earlier work in prompt optimization [1] reveals that the demonstrations serve as a key component in prompt design for performance improvement, which has been further validated in [2] that explicitly explains why prompt optimization helps so much. It provides empirical evidence that the optimized prompt can guide agents to imitate successful reasoning patterns and follow a strict format via in-context learning [2]. Better instruction design can also steer LLM agents towards strategic reasoning patterns. In our work, the importance of prompts is amplified due to the multi-agent cooperation, where errors can even propagate through the chain of agents. We refer to the detailed prompts and optimized examples in Appendix F. For instance, in MBPP, the optimized prompt involves detailed format controls and Python example scripts, which enforce better formats for language agents and largely reduce error rates. We thank the reviewer for raising this discussion and will highlight it with optimized prompts
> >
> > [1] Opsahl-Ong, K., Ryan, M. J., Purtell, J., Broman, D., Potts, C., Zaharia, M., & Khattab, O. (2024). Optimizing Instructions and Demonstrations for Multi-Stage Language Model Programs. EMNLP 2024.
> >
> > [2] Wan, X., Sun, R., Nakhost, H., & Arik, S. O. (2024). Teach Better or Show Smarter? On Instructions and Exemplars in Automatic Prompt Optimization. NeurIPS 2024.
> >
> > > 4.2. Topology analysis: Fig 4 shows different topologies work for different tasks, but there's limited discussion of why. What task characteristics favor which topologies?
> >
> > We kindly refer the reviewer to Appendix D.2, where we explicitly discussed the MASS-found topologies and characteristics of these topologies. We show that the optimized topology exhibits certain patterns per task family. For example, the “debate” topology is a winning pattern across all multi-hop reasoning tasks for eliciting more truthful predictions, where MASS can automatically identify the most influential topology from any customized search space.
> >
> > > 4.3. Workflow-level optimization gains: Stage 3 provides only ~2% improvement (line 342). Is this consistent? When does it help?
> >
> > In Table 6, Appendix D, we provide the detailed ablation results per optimization stage of MASS. It is observable that 3PO consistently brings marginal but practical gains over 2TO, and the MAS performance becomes saturated after 3PO, achieving state-of-the-art performance with a signal of convergence. The gain of 3PO comes from optimizing the interdependence across agents through better prompt design, which leads to improved global performance when 1PO->2TO is not saturated.

---

### Author Response · Authors · 2025-11-22
**Overall response**

We sincerely thank all reviewers for their insightful feedback and engagement during the discussion period, which has helped us to strengthen our manuscript. We are grateful that reviewer ib6h found our response convincing and raised their score after the initial response. We are glad that all reviewers consistently find our approach intuitive and well-motivated, commonly acknowledge the novelty of MASS, and the extensive and strong empirical results.

## Key Changes

We believe that we have responded to each of the questions from reviewers thoroughly in the rebuttal. In light of the comments from reviewers, we summarize the key changes in the latest revision. We highlight all changes and additions in Magenta for the convenience of the reviewers, and below we list all key changes:

- Updated the notations in Section 2 to clarify the definition of $a$ and its connection to individual agents and topologies.
- Updated Section 3.2 with a clarification of notations on search dimensions.
- Moved the Related Work section to Section 5 for a better logical flow.
- Corrected typo in Appendix C.2.
- Added a cost analysis with the scalability of MASS in Appendix D.3.

---

### Meta-Review · Area_Chair_S7kw · 2026-01-02

**Summary:**

Reviewers mainly focused on the perceived level of novelty of the proposed framework, the soundness and fairness of the experimental methodology (including search space design, pruning strategy, baseline comparisons, and statistical validity), as well as clarity and presentation issues such as notation consistency and paper organization. Overall, concerns were centered on whether the three-stage design was sufficiently justified and whether the empirical analysis was thorough and transparent enough to support the claimed advantages.

**Reviewer Concerns:**

Most substantive concerns were addressed in the rebuttal and revision. The authors clarified ambiguous notations, improved presentation, added statistical significance tests and scalability analysis, and provided additional experiments and explanations regarding prompt-first versus topology-first optimization and search space pruning. These responses resolved the majority of methodological and clarity-related issues. Some concerns regarding the breadth of the topology search space, fixed module ordering, and the limited theoretical grounding remain partially open, but are largely framed as design trade-offs or future research directions rather than critical flaws.

**Reviewer Scores:**

Following the rebuttal, reviewers who raised clarity and methodological concerns are expected to moderately increase their scores, with at least one reviewer explicitly indicating a score raise. Other reviewers are likely to maintain or slightly improve their original scores, as key weaknesses were addressed while residual concerns mainly reflect differing expectations on novelty and theory. Overall, the post-rebuttal sentiment trends modestly more positive.

---

### Decision · Program_Chairs · 2026-01-26

Accept (Poster)